# Construction of a trio-based structural variation panel utilizing activated T lymphocytes and long-read sequencing technology

Akihito Otsuki [1,2,7], Yasunobu Okamura [1,3,7], Noriko Ishida[1], Shu Tadaka [1], Jun Takayama[1,3,4,5], Kazuki Kumada[1], Junko Kawashima[1], Keiko Taguchi [1,2,3], Naoko Minegishi[1], Shinichi Kuriyama[1], Gen Tamiya[1,3,4,5], Kengo Kinoshita[1,2,3,6], Fumiki Katsuoka [1,3] & Masayuki Yamamoto [1,2,3 ✉]

Long-read sequencing technology enable better characterization of structural variants (SVs). To adapt the technology to population-scale analyses, one critical issue is to obtain sufficient amount of high-molecular-weight genomic DNA. Here, we propose utilizing activated T lymphocytes, which can be established efficiently in a biobank to stably supply high-grade genomic DNA sufficiently. We conducted nanopore sequencing of 333 individuals constituting 111 trios with high-coverage long-read sequencing data (depth 22.2x, N50 of 25.8 kb) and identified 74,201 SVs. Our trio-based analysis revealed that more than 95% of the SVs were concordant with Mendelian inheritance. We also identified SVs associated with clinical phenotypes, all of which appear to be stably transmitted from parents to offspring. Our data provide a catalog of SVs in the general Japanese population, and the applied approach using the activated T-lymphocyte resource will contribute to biobank-based human genetic studies focusing on SVs at the population scale.

[1] Tohoku Medical Megabank Organization, Tohoku University, 2-1 Seiryo-machi, Aoba-ku, Sendai, Miyagi 980-8573, Japan. [2] Department of Medical Biochemistry, Tohoku University Graduate School of Medicine, 2-1 Seiryo-machi, Aoba-ku, Sendai, Miyagi 980-8575, Japan. [3] Advanced Research Center for Innovations in Next-Generation Medicine, Tohoku University, 2-1 Seiryo-machi, Aoba-ku, Sendai, Miyagi 980-8573, Japan. [4] Statistical Genetics Team, RIKEN Center for Advanced Intelligence Project, Nihonbashi 1-chome Mitsui Building 15 F, 1-4-1 Nihonbashi, Chuo-ku, Tokyo 103-0027, Japan. [5] Department of AI and Innovative Medicine, Tohoku University Graduate School of Medicine, 2-1 Seiryo-machi, Aoba-ku, Sendai, Miyagi 980-8575, Japan. [6] Graduate School of Information Sciences, Tohoku University, 6-3-09 Aramaki Aza-Aoba, Aoba-ku, Sendai, Miyagi 980-8579, Japan. [7] These authors contributed equally: Akihito Otsuki, Yasunobu Okamura. ✉email: masiyamamoto@med.tohoku.ac.jp

To realize and facilitate genomic medicine, it is essential to identify and understand the genetic variations present in the general population. Over the last decade, population-scale genome studies have been conducted in many countries, providing insight into the processes involved in genetic diversity and disease biology[1–3]. Population allele frequencies have been actively used for the diagnosis of rare diseases as a reference panel[4–7]. A set of phased variants in individual haplotypes have been utilized to design DNA arrays[8] and for genotype imputation[9–11].

To enhance genome research in the Japanese population, we established the Tohoku Medical Megabank (TMM) Project consisting of a community-based cohort[12] and birth and three-generation cohort (BirThree Cohort)[13] with a biobank (TMM Biobank)[14]. We constructed an allele-frequency panel focusing on single-nucleotide variants (SNVs) and short insertions and deletions (indels) based on short-read whole-genome sequencing (WGS) analyses. The current panel covers more than 38,000 participants[5,15–17]. These data have been prepared along with multi-omics data and medical information, and they have become a useful resource for academic, clinical, and industrial research[17,18]. However, a reference panel focusing on structural variants (SVs) in the Japanese population is lacking.

SVs are typically defined as genomic rearrangements of more than 50 bp in size[19–21]. SVs are more likely to be associated with a genome-wide association signal and affect gene expression[20,22], because larger changes in the genome are likely to result in deleterious events through the disruption of protein synthesis and the regulatory mechanisms of gene expression[23].

Despite the biological importance of SVs, a limited number of large-scale genome studies have focused on SVs at the population scale[24–27] due to the fundamental limitations of conventional short-read sequencing technologies in SV detection. In contrast, recent developments in long-read sequencing technologies have led to new approaches for genome analysis, such as nanopore sequencing by Oxford Nanopore Technologies[28,29], and single-molecule real-time sequencing by Pacific Biosciences[30,31]. These long-read sequencing technologies enable us to sequence several thousand base pairs or more, which are more likely to span the breakpoints of SVs with high-confidence alignments, aiding in the capture of larger SVs better than can be achieved with short-read sequencing alone[32–34].

Pioneer studies have taken advantage of such long-read technologies for SV analysis, giving rise to new challenges in population-scale studies[35,36]. In particular, it is preferable to prepare a relatively large amount of high-quality genomic DNA, as the data derived from a long-read sequencing analysis are affected by various factors[37]. For instance, the molecular weight of genomic DNA has been shown to affect the sequencing yield[35], which seems to influence downstream variant detection, resulting in the under- or overestimation of allele frequencies. Population-scale studies involving long-read sequencing technologies have been addressing these issues[35,36], but the problems of a nonuniform distribution of read lengths and low sequencing depth coverage remain.

To overcome these hurdles inherent to population-scale long-read sequencing analyses and construct a Japanese-specific allele-frequency panel focusing on SVs, we employed a unique cell resource of TMM Biobank. To date, we generated more than four thousand Epstein–Barr virus (EBV)-transformed lymphoblastoid cell lines (LCLs) and activated T cells, which have been utilized in the genome and functional studies[38,39]. In this study, we utilized the activated T cells as a genomic DNA resource and conducted the long-read sequencing analysis of 333 Japanese individuals constituting 111 trios, who were recruited in the TMM BirThree Cohort Project. We cataloged the allele frequencies of 72,470 SVs located in autosomes. Thus, we succeeded in constructing a Japanese population-scale SV panel of Japanese, providing a fundamental resource for human genetic studies.

## Results

**Activated T cells as a DNA resource for long-read sequencing analyses.** To capitalize on the advantage of long-read sequence technologies in SV analyses, high-molecular-weight DNA should be used for the library preparation step because the read length depends on the DNA size of DNA fragments in a library. Similarly, the data yield for a given cost varies depending on the input. For instance, it has been implied that the fragmentation status affects the sequencing yield per run, indicating that the length of the input libraries can affect the sequencing depth per cost[35]. As the read length and sequencing depth are important factors in constructing variation panels, these factors should be carefully controlled. In this regard, the TMM Biobank has been establishing proliferating cell resources, which avoid the rapid depletion of biospecimens[14]. In this study, we used high-molecular-weight DNA specimens derived from activated T lymphocytes for long-read sequence analyses.

The activated T cells were established from CD19-negative cells from the PBMC fraction in the blood of participants (Fig. 1a). As of March 2021, the TMM Biobank had established 4527 LCLs and 4808 activated T lymphocytes (Fig. 1b). The reason for our selection of genomic DNA specimens derived from activated T cells versus LCLs for the long-read sequence analysis is that the former can be established in a much shorter time span with a higher success rate than the latter, ensuring future expandability. We stimulated the cells with the human T-cell activators CD3 and CD28 and expanded them for three to ten days in culture medium supplemented with recombinant IL-2 cytokine[14]. We successfully recovered more than 99% of the cells stored in liquid nitrogen. Most of these cell resources are accompanied WGS information determined by short-read sequencing. Almost all established cells were positive for CD3, a T-cell marker (Fig. 1c), indicating that T cells dominantly proliferated under cytokine stimulation.

To assess the quality of the genomic DNA samples extracted from these T cells, we measured the optical density (OD) at 260/280 and 260/230 and obtained average OD ratios of $1.89 \pm 0.15$ and $1.81 \pm 0.44$ (mean ± SD), respectively. We also conducted pulsed-field gel electrophoresis of 5 random samples (#1–#5). The lengths of the DNA specimens ranged from 20 to 145 kb (Fig. 1d and Supplementary Fig. 1a), demonstrating that the DNA samples used were appropriate for long-read sequencing analyses.

Next, we conducted a long-read WGS analysis with a nanopore sequencer utilizing T-cell genomic DNA specimens. To optimize the DNA fragmentation step to balance the sequencing yield and read length, we designed a step to yield DNA fragments with lengths ranging from 20 to 80 kb (Fig. 1d and Supplementary Fig. 1a). We obtained $85.0 \pm 5.4$ Gb of yield and $25.8 \pm 1.8$ kb of N50 length per flowcell ($n = 5$), indicating that half of the sequence base pairs were derived from reads longer than or equal to 25.8 kb (Fig. 1e and Supplementary Fig. 1b, c). Taken together, these results support our contention that activated T cells constitute a useful genomic DNA resource for a long-read WGS analysis.

To address whether the utilization of activated T cells is a good method for genome analyses, we designed a benchmark analysis using three independent sets of genomic DNA samples obtained from activated T cells and LCLs (Supplementary Fig. 2a). Notably, there are donor-matched and high-quality de novo assemblies available for all three genomes[40]. In this benchmark analysis, we obtained standard SV-call sets from the assemblies and compared them to SV-call sets from nanopore sequencing data to calculate the precision and recall scores of SV detection (Supplementary Fig. 2a).

To select the SV-call pipeline, we experimentally compared the efficiency and accuracy of the CuteSV[41] and Sniffles algorithms[42], both of which are widely used algorithms available for the nanopore sequencing data. As shown in Supplementary Fig. 2b, CuteSV reproducibly showed higher recall and precision scores than Sniffles in the detection of DELs. This result is concordant with previous benchmark studies[41,43], and we therefore decided to utilize the CuteSV algorithm. Utilizing this algorithm, we next compared the precision and recall scores of activated T cells and LCLs. As shown in Supplementary Fig. 2c, we observed that activated T cells and LCLs showed very similar recall and precision scores. Therefore, we concluded that activated T cells were an acceptable resource for a genome analysis similar to LCLs.

**Trio-based structural variation analysis using long-read sequencing technology**. To clarify the variation spectra, frequencies, and functional impact of SVs in the Japanese population, we constructed an allele-frequency panel. Here, it should be

noted that despite the continuous improvements in computational tools, many challenges in read alignment-based SV calling algorithms remain[34,41]. Therefore, to apply quality assessments based on Mendelian inheritance error profiling, we designed WGS analyses of 333 BirThree cohort participants comprising 111 parent–offspring trios through the long-read sequence procedures established in this study (Fig. 2a).

Using 411 flowcells, we conducted 430 runs in total (Supplementary Data 1). As shown in Supplementary Fig. 3a, b, the sequencing yields increased as the active pore count increased, but the N50 lengths did not. These observations indicate that the quality of the flowcell is a determinant of the sequencing yield. While it has been known that the sequencing yield per flowcell decreases when longer libraries are subjected to sequencing[35], we did not observe such a negative correlation between the two variables (Supplementary Fig. 3c). We surmise that this occurred because optimization in the fragmentation step resulted in low diversity of the N50 length, suggesting that it is important to optimize the DNA fragmentation step to balance the sequencing yield and read length for high-quality deep sequencing.

The sequencing data resulted in 69.7 Gb per individual after filtering the sequence reads with low-quality values (lower than a mean quality score[44,45] of 6 as shown by the dotted line in Fig. 1e). Our strategy led to relatively long (read N50 of $25.8 \pm 3.9$ kb) sequence reads (Fig. 2b) compared to previous works[35,36]. When aligned to the human reference genome (GRCh38), the sequence reads resulted in $22.2 \pm 4.4$-fold coverage (Fig. 2c), and the median sequencing error rate was 7.9% (2.2% for insertions, 3.5% for deletions, and 2.2% for mismatches) (Supplementary Fig. 3d). Taken together, the results support the integrity of our approach, including the following two important improvements: the use of T-cell resources to stably provide high-quality DNA suitable for SV analyses at the population scale and the use of BirThree Cohort participants for Mendelian error profiling.

**Structural variations detected in the Japanese population**. Next, we evaluated the structural variation spectra in the Japanese population. We detected two classes of canonical SVs, deletions (DELs) and insertions (INSs), both of which were more than 50 bp in length, identifying $23,056 \pm 454$ SVs per individual on autosomes composed of 10,923 DELs and 12,133 INSs per individual (Fig. 2d). The numbers of detected SVs in this study are comparable to those of previous estimations by means of long-read sequencing[32,35] and other technologies[21,33].

Then, we merged these SVs of the 333 individuals into a nonredundant set of SVs to produce a variant repository composed of 37,981 DELs and 36,220 INSs, showing a balanced number of DELs and INSs (Fig. 2e). In this regard, several studies have identified more INSs than DELs[35,46]. A plausible explanation for this discrepancy may be the lower recall scores in INS detection than DEL detection in our study (Supplementary Fig. 2b). We surmise that biases in SV calling remain and expect the development of elaborate bioinformatics algorithms to address this issue. One additional hypothesis is that a few thousand loci that are underrepresented in GRCh38 may affect the ratio between insertions and deletions. In good agreement with this hypothesis, while we identified more insertions than deletions in the individual-level analysis (Fig. 2d), the ratio between insertions and deletions became closer to 50:50 in the population-scale analysis that included 333 individuals (Fig. 2e).

The number of SVs strongly correlates with the length and rapidly decreases; three peaks at sizes of ~300 bp, 3 kb, and 6 kb are notable. We consider that these peaks are due to retrotransposon elements, especially Alu, SINE/VNTR/Alu, and LINE-1 elements (Fig. 2e), based on their sizes[35,36].

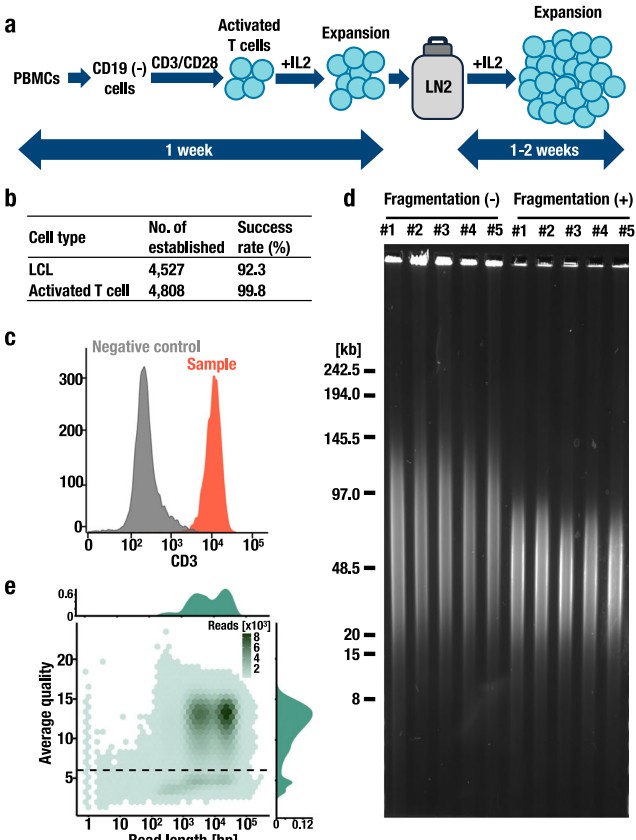

**Fig. 1 Long-read sequencing using activated T cells. a** Scheme of the establishment of activated T cells. T cells were established from CD19-negative cells derived from PBMCs by CD3/CD28 stimulation and stored in liquid nitrogen. After freezing and thawing, the cells were expanded under IL-2 stimulation. **b** Numbers of cell resources established in the TMM Biobank. Success rates of the establishment processes are also shown. **c** Cell surface marker profiles of activated T lymphocytes. **d** Length of genomic DNA as assessed by pulsed-field gel electrophoresis. Genomic DNA isolated from activated T cells was fragmented using a 29-gauge needle and syringe pump. Representative images of five independent samples (from #1 to #5) before (−) and after (+) the fragmentation steps are shown. **e** Bivariate plot of the read length (x axis) and aligned read quality (y axis) with kernel density estimation. The threshold used to filter low-quality sequence reads (mean quality score of 6) is shown as a dotted line.

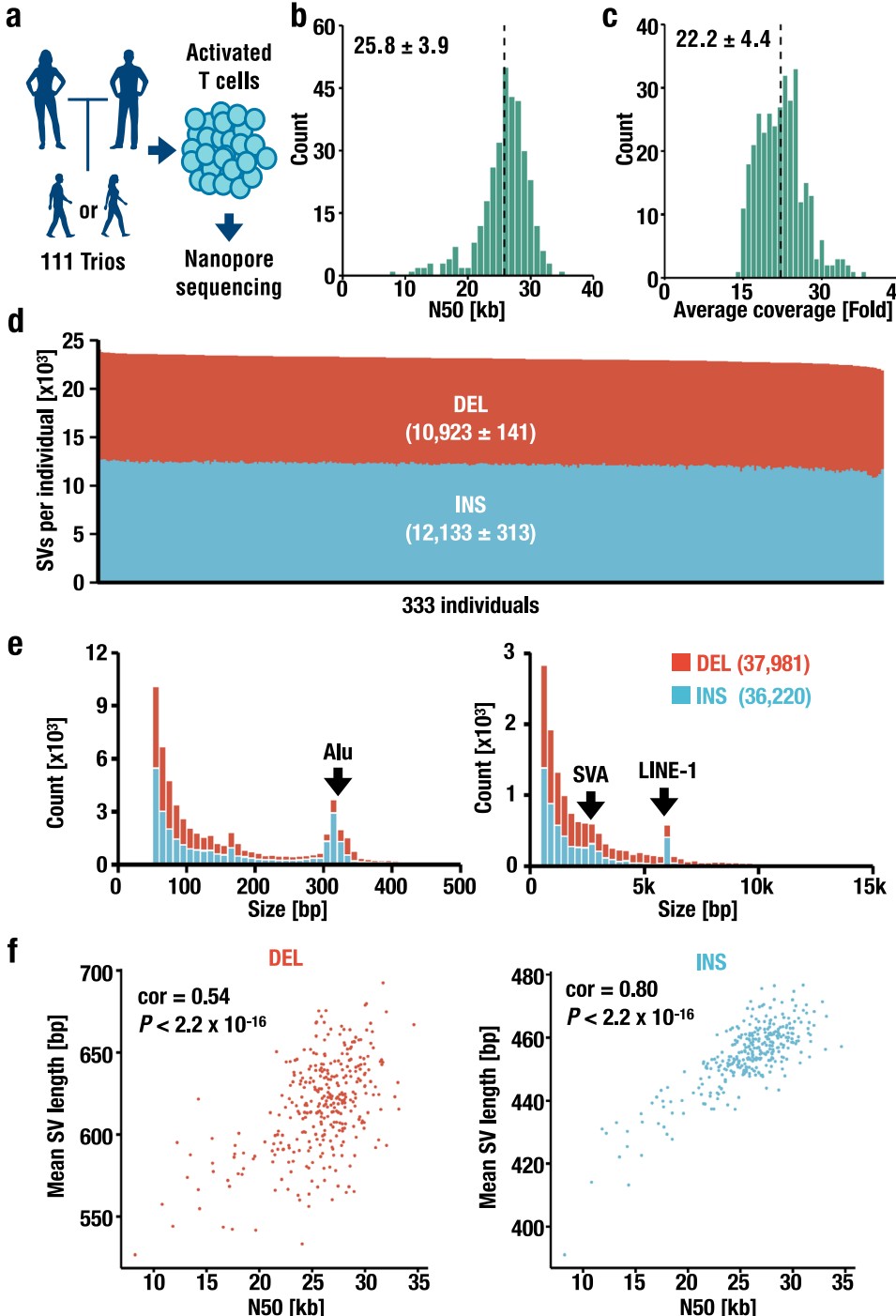

**Fig. 2 Study design and statistics of the trio-based nanopore sequencing. a** Design of the SV analysis in this study. Activated T cells were established from 333 individuals comprising 111 trios and subjected to whole-genome sequencing using a nanopore sequencer. **b, c** Distribution of read length N50 (**b**) and average coverage of aligned reads (**c**). The dotted lines show the mean values, and the mean ± SD is indicated in the panel. **d** Number of DEL (red) and INS (blue) SVs located on autosomes per individual. Mean ± SD is indicated in parentheses. **e** Size distribution of the DELs (red) and INSs (blue). The left and right panels indicate the distribution of SVs with sizes from 50 to 500 bp (bin = 10 bp) and sizes from 500 bp to 15 kb (bin = 300 bp), respectively. Notable peaks due to transposable elements (Alu, SINE/VNTR/Alu [SVA], and LINE-1) are shown as arrows. The numbers of DELs (red) and INSs (blue) are shown in parentheses. **f** Scatterplot showing the relationship between the N50 length and mean DEL (red) and INS (blue) length. The Pearson correlation coefficient (cor) and P values are shown (n = 333).

To ascertain the benefit of using high-quality DNA in SV analyses, we examined the correlation between the read length and SV detection ability. As shown in Fig. 2f, we observed a strong correlation between the read N50 and the mean size of the INSs and a moderate correlation between the read N50 and the mean size of the DELs (Pearson correlation coefficient [cor] = 0.54 and 0.80 for DEL and INS, respectively). These results suggest that an SV analysis using longer reads has an advantage in the detection of large SVs compared with that using shorter reads. To further verify this finding, we also evaluated the correlation between the read length and SV

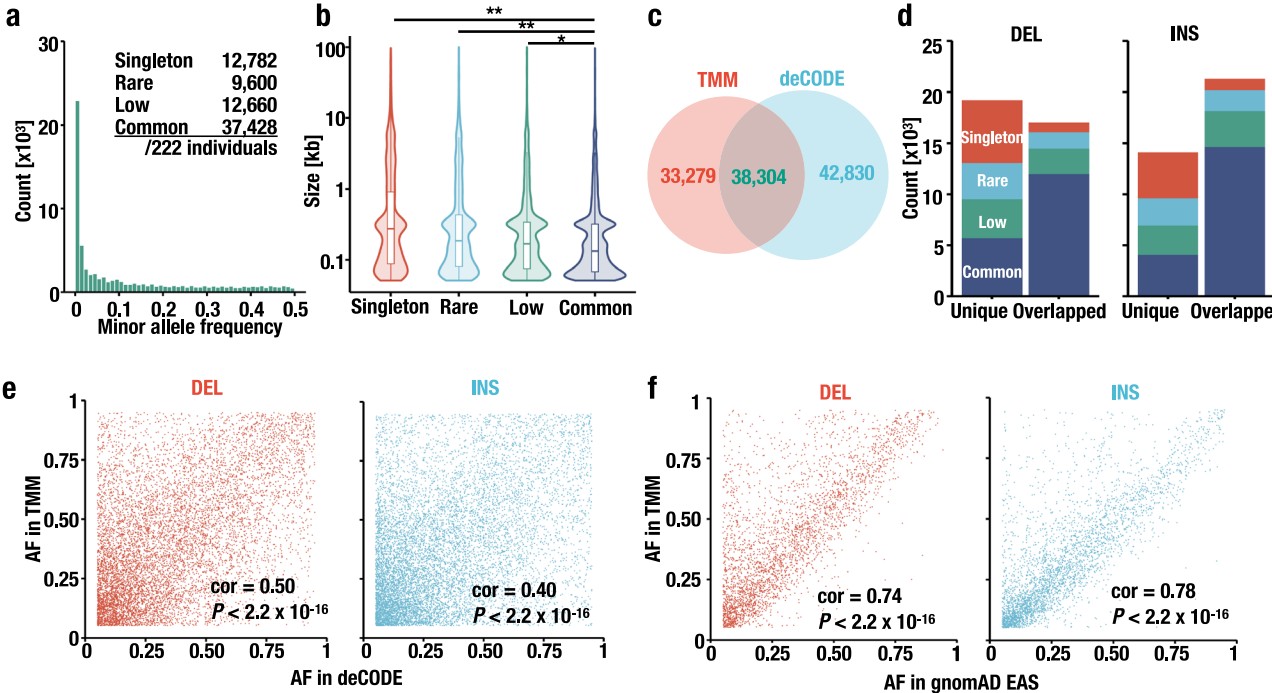

**Fig. 3 Characteristics of SVs detected in the Japanese population. a** Distribution of minor allele frequency (MAF). SVs were categorized as follows: singleton (minor allele count [MAC] = 1), rare (MAC > 1 and MAF < 1%), low (MAF ≥ 1% but <5%), and common (MAF ≥ 5%). **b** Distribution of the SV size in each MAF category. *$P = 0.00098$ and **$P < 2 × 10^{-16}$, Holm adjusted Wilcoxon rank-sum test. The numbers of SVs belonging to each category are shown in Fig. 3a. Each boxplot has a box that represents the interquartile range (IQR) and whiskers that extend 1.5 × IQR from the box edges. The median is shown in a horizontal line in the box. **c** Overlap between our dataset and a previous report[35]. The numbers of SVs identified in this work (TMM), and previous work (deCODE) are shown in red and blue, respectively. The number of SVs identified in both studies is shown in green. **d** Bar plot showing the proportion of MAF categories in DELs (left) and INSs (right) identified in the TMM dataset only (unique) and in both the TMM and deCODE datasets (overlapped). **e** Scatterplot showing the allele frequencies estimated in the deCODE study (x axis) and this study (y axis). The Pearson correlation coefficient (cor) and P values are shown. $n = 8568$ and 9762 for DEL and INS, respectively. **f** Scatterplot showing the allele frequencies estimated in the East Asian subset of the gnomAD study (x axis) and this study (y axis). The Pearson correlation coefficient (cor) and P values are shown. $n = 3491$ and 3515 for DEL and INS, respectively.

detection ability by detecting large SVs ranging from 5.9 to 6.1 kb and small SVs ranging from 280 to 350 bp (Supplementary Fig. 4a, b, respectively). The SVs belonging to the former fraction contain LINE-1-related SVs, and those belonging to the latter fraction contain Alu-related SVs. We found that the number of large INSs was correlated with the read N50 (cor = 0.54), whereas that of the DELs (cor = 0.18 and 0.07 for large and small DELs, respectively) and small INSs (cor = 0.25) was not correlated with the read N50. These results indicate that an SV analysis using longer reads ranging from 10 to 35 kb utilizing activated T cells is beneficial for the comprehensive identification of large SVs, especially in the case of INSs.

Next, we evaluated the minor allele frequencies (MAFs) of individual SVs in the Japanese population. To avoid double counting the SVs shared between parents and offspring and, thus, prevent the overestimation of the allele frequencies of the SVs in the general population, we extracted SVs observed in 222 unrelated individuals (i.e., fathers and mothers) from the repository to evaluate MAF. We found that the number of SVs decreased as the allele frequency increased (Fig. 3a). Then, we categorized these SVs into the following four categories: singleton (minor allele count [MAC] = 1); rare (MAC > 1 and MAF < 0.01); low (MAF ≥ 0.01 and MAF < 0.05); and common (MAF ≥ 0.05). Across all SV classes, 12,782 SVs (representing 17.6% of all SVs identified in 222 unrelated individuals) were singletons, 9600 (13.2%) SVs were rare, 12,660 (17.5%) SVs were low, and 37,428 (51.6%) SVs were common. Overall, most SVs are shared among unrelated individuals.

An intriguing observation is that the sizes of the SVs vary among the SV categories (Fig. 3b). For example, large SVs were

most frequently found in the singleton category, but the median size of the SVs decreased as the MAF increased (singleton, 274 bp; rare, 201 bp; low, 168 bp; and common, 133 bp). This result is consistent with the previous observation[47] in which the allele frequency of SVs in size range of 100 kb to 1 Mb decreased with size. The size of the SVs appeared to be the smallest in the common category. These observations suggest that the size of SVs or the amount of rearranged DNA may be a key determinant in the selection of SVs.

**Ethnic diversity of SVs**. To assess ethnic differences or diversity in the occurrence of SVs, we compared the DELs in our dataset with those in the recently published Iceland deCODE study[35] (see Methods, "Comparison of SVs to the deCODE dataset"). The deCODE dataset contains data derived from a population-based analysis of SVs using a long-read sequencing platform. As shown in Fig. 3c, of all SVs in our dataset (INS and DEL; shown as "TMM"), 38,304 (53.5%) were also found in the deCODE dataset, while 33,279 were unique to the TMM dataset. Next, we compared the MAFs of the unique DELs and INSs to those of overlapping ones. The results revealed that the SVs in the common category were shared preferentially with those in the deCODE dataset; in contrast, those with a low MAF and those in the rare and singleton categories tended to be unique in the TMM dataset (Fig. 3d). Thus, the comparison of the deCODE and TMM datasets revealed significant differences in the ethnic distribution of SVs, even though high-MAF SVs are shared relatively widely across ethnicities.

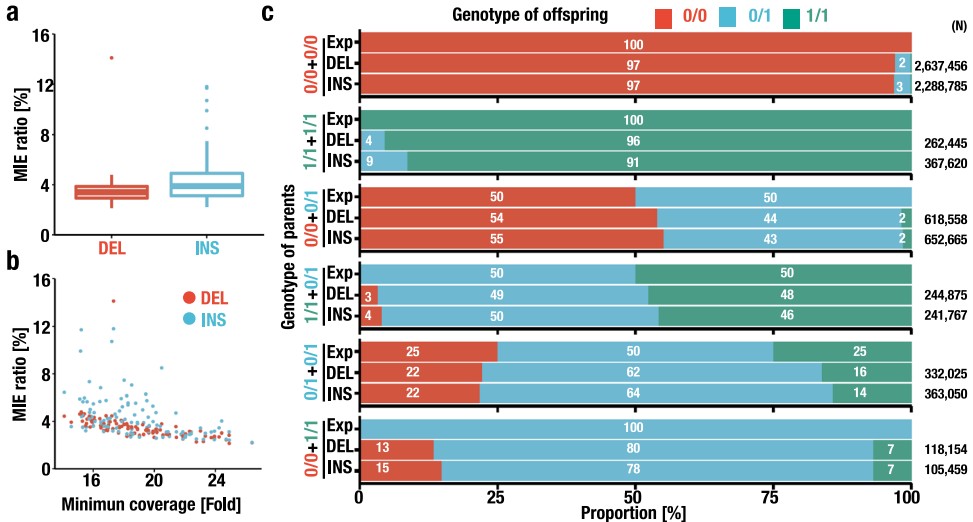

**Fig. 4 Quality assessment of the SV dataset based on Mendelian inheritance. a** Mendelian inheritance error (MIE) ratio per trio. The MIE ratio was calculated for each trio. Each boxplot has a box that represents the interquartile range (IQR) and whiskers that extend 1.5 × IQR from the box edges. The median is shown in a horizontal line in the box. Outliers are shown in dots. $n = 111$ for both DEL and INS. **b** Scatterplot showing the relationship between the sequencing coverage and the MIE ratio. The minimum coverages among three individuals who belong to a trio and the MIE ratio observed in the trio are plotted on the x- and y- axes, respectively. $n = 111$ for both DEL and INS. **c** Distribution of genotype calls in parent–offspring trios. 0 and 1 denote the reference and alternative alleles, respectively. Genotypes of the parents are shown on the y axis, and the frequencies of each genotype (0/0, 0/1, and 1/1; red, blue, and green, respectively) that were expected (Exp) and observed in DELs (DEL) and INSs (INS) are shown. The number of pairs tested are shown on the right.

For further analysis focusing on the ethnic diversity of SVs, we extracted common SVs (MAF ≥ 0.05) and compared the allele frequencies (AFs) between the deCODE dataset and the TMM dataset (Fig. 3e). We observed only modest correlations between these two datasets (Pearson correlation coefficient [cor] = 0.50 and 0.40 for DEL and INS, respectively), indicating that the common SVs showed substantial differences in AFs between the ethnicities.

Next, we compared the TMM dataset to that of gnomAD-SV[25], which was based on short-read WGS of more than 14,000 samples. Notably, the samples are mainly derived from European and African/African-American samples, while less than 10% of the samples are derived from East Asian samples (EAS). In the comparison between the TMM dataset and the whole gnomAD-SV, only modest correlations (cor = 0.61 and 0.57 for DELs and INSs, respectively) were observed (Supplementary Fig. 5), showing very good agreement with the observation in the comparison to the deCODE dataset. In contrast, we observed much higher correlations (cor = 0.74 and 0.78 for DELs and INSs, respectively) in the comparison of the TMM dataset and the East Asian subset in the gnomAD-SV (Fig. 3f), demonstrating that the AFs in the TMM dataset reflect ethnicities in AFs.

**Quality assessment of SVs based on Mendelian inheritance.** To assess the reliability of our SV analysis and dataset, we examined the Mendelian inheritance error (MIE) rates by taking advantage of a trio analysis. In general, MIEs arise from two major events in this type of analysis. One event is germline or nongermline de novo mutations (biological errors), and the other event is variant calling errors or incorrect pedigree information (technical errors). As true de novo mutations occur at an extremely low rate ($\sim 1 \times 10^{-8}$ per base pair per generation)[48,49], the former biological errors seem less influential in this case, and we surmise that most MIEs are the consequence of the latter event. Therefore, the MIEs of SVs can serve as an indicator of the reliability of SV analyses. In fact, the MIE rate was calculated in the deCODE long-read study[35].

We found that the transmission of 3.5 ± 0.1% and 4.3 ± 0.2% of the DELs and INSs per trio, respectively, did not follow Mendelian inheritance (Fig. 4a). We also observed a lower concordance in trios with lower coverage (Fig. 4b). In particular, the genotyping accuracy of the INSs was more sensitive to the sequencing coverage, indicating that a higher sequencing coverage is desirable for higher genotyping accuracy. A previous analysis of five trios (15 members) using long-read sequencing technology estimated the MIE to be 6.4–15.2%[50]. Thus, although the MIE rates were still higher than expected, there was a substantial improvement in concordance with Mendelian inheritance, perhaps due to technical improvements in long-read sequencing, including stable data production in terms of the sequencing depth and read length, and in bioinformatics pipelines.

We attempted to examine the characteristics and genomic locations of the SVs that showed MIEs. Taking advantage of the trio analysis, we evaluated the incidence of MIEs in each SV by calculating the error family ratio (number of trios with MIE/total number of trios analyzed). For the SV types that showed MIEs, we observed concordant values in the DELs and INSs (DEL, 3.2 ± 5.3%; INS, 3.7 ± 5.2% of trios showed MIEs; Supplementary Fig. 6a). To address the genomic distribution of SVs with MIEs, we precisely plotted the error family ratios on genomic locations (Supplementary Fig. 6b). We identified SVs located near gaps and chromosome ends that frequently accompany MIEs. We also found that the SVs in the regions near gaps and chromosome ends were often called based on low coverage sequencing reads (Supplementary Fig. 7a, b). Therefore, we surmise that the high incidences of MIEs were derived, at least partly, from erroneous SV calls due to the difficulty in read mapping.

We also evaluated the accuracy of the SV in each MAF category. To this end, we employed the error family ratio. As shown in Supplementary Fig. 6c, we observed that the error family ratios were lower for singleton SVs than for common or highly frequent SVs. One plausible explanation for this unexpected observation is that for high-frequency SVs, the accuracy is affected by systematic errors during the identification of SVs. In low complexity regions in the genome, some SVs with a

low accuracy may be called with high frequencies, which results in MIEs. In contrast, singletons and low-frequency SVs may be less affected by such systematic errors.

We evaluated the distribution of the SV genotype calls in all parent–offspring trios to assess whether there are specific combinations of genotypes among trios that lead to erroneous calls (Fig. 4c). While the distributions of DELs and INSs in our dataset were closer to the expected probabilities, for the most part, several minor discrepancies in the genotype distribution were observed. For instance, in the case of the genotype of offspring derived from parents with the genotypes "0/0" and "1/1", high-rate MIE accumulation was observed (Fig. 4c). This observation was reproducible in the deCODE study[35]. Thus, these results demonstrate that our long-read sequence analysis achieved reasonable genotype calls, considering that substantial challenges need to be addressed to accomplish fully reliable and accurate genotype calls of SVs using long-read sequence technology.

**Functional annotation of SVs**. To evaluate the potential function of the SVs, next, we annotated the SVs to genomic features. We first examined how these SVs are distributed on chromosomes. Although the number of SVs correlated with the chromosome length (Fig. 5a), the SVs were differentially distributed on each chromosome (Fig. 5b; $P < 2.2 \times 10^{-16}$ for DELs and $P = 6.87 \times 10^{-16}$ for INSs; Kruskal–Wallis rank-sum test). We also observed several peaks, indicating the high density of SVs on chromosome ends and sites adjacent to the gaps remaining in the reference genome, and this observation is concordant with the observation in a previous study[32] (Fig. 5c). To evaluate the nonrandom distribution of the SVs in detail, we plotted the number of sequencing reads supporting the variant call of each SV (Supplementary Fig. 7a, b). The sequencing depths were decreased in the genomic regions adjacent to gap and chromosome ends, suggesting that the difficulty in read mapping might result in the erroneous detection of SVs in these regions. Nonetheless, we found five peaks of SVs located in positions far from the gaps and chromosome ends (Fig. 5c, green arrows). The green arrow position on chromosome 6 involves human leukocyte antigen (HLA) loci. Regarding the position, a closer analysis of our long-read sequence data revealed that these five peaks are located in regions that harbor high-level segmental duplications (Fig. 5c). The regions with SD accumulations are intractable with the current long-read sequencing technology. We surmise that the difficulty in SV detection in these regions might result in the overestimation of SVs.

Next, we examined the localization of these SVs within intergenic regions, introns, exons, and protein-coding sequences (CDSs). Of 74,201 SVs, 34,053 (45.9%) were in intergenic regions, while 38,749 (52.2%), 3099 (4.2%), and 828 (1.1%) SVs overlapped with introns, exons, and CDSs, respectively ($P < 0.002$, bootstrap test), which is concordant with a previous study[35]. Thus, SVs located in intergenic regions were over-represented, and SVs located in introns, exons, and CDSs were underrepresented (Fig. 5d). We also observed elevated rates of rare alleles in exons and CDSs (Fig. 5e). These differences in the distribution of SVs within genomic regions suggest that these SVs influence gene structure, which provides information regarding the strength of negative selection during molecular evolution.

**SVs associated with clinical phenotypes**. As SVs affect gene structures more drastically than smaller variants, including SNVs and indels (insertions/deletions less than 50 bp), SVs located in CDSs may exert more potent effects on gene functions and downstream phenotypes than SNVs and indels. To gain functional insight into these SVs, we searched for and identified SVs overlapping with CDSs belonging to 461 protein-coding genes (Supplementary Data 2). Of these SVs, we selected four

previously shown or suggested to be associated with clinical phenotypes and examined them closely in our set of analyses.

We identified DELs of 4.9 kb in hemoglobin subunit gamma 1 and 2 (*HBG1* and *HBG2*) loci (Fig. 6a), which were present in two parent-offspring pairs out of 111 trios examined in this study (Fig. 6f), and in both cases, the alleles transmitted from parent to offspring (from the father to offspring in the family shown in Fig. 6a). While the *HBG1* and *HBG2* genes, which encode γ-globin chains consisting of HbF, are expressed predominantly during the fetal stage from the β-globin gene cluster, their expression is progressively silenced during the postnatal period due to the interplay of transcription factors interacting with the locus control region (LCR) and the *HBG1* and *HBG2* promoters[51]. The breakpoints of the DELs are located in second introns of the *HBG1* and *HBG2* genes, resulting in an *HBG1*-*HBG2* fusion gene[52], as illustrated in Fig. 6e. Intriguingly, this DEL has been shown to cause an increased expression of γ-globin and elicit hereditary persistence of fetal globin (HPFH), with increased expression of γ-globin in the adult stage[52]. Another East Asian case of HPFH with this DEL has also been reported[53]. The allele frequency (AF) of this DEL was 0.45% (2 in 444 alleles) in this study, which appears to be concordant with the estimation obtained using a short-read whole-genome sequence database (Fig. 6f; 0.52% in East Asia from gnomAD[25]).

We also found a 32-kb DEL in genes encoding late cornified envelope 3B and 3C (*LCE3B* and *LCE3C*) proteins related to skin barrier functions, as has been described in the previous studies[35,54]. This DEL includes whole *LCE3B* and *LCE3C* genes (Fig. 6b), and complete loss of these genes is reported to be associated with susceptibility to psoriasis[35,55,56]. The AF of this DEL was 49.3% in our study, showing very good agreement with the high frequency in East Asia (57.7%; Fig. 6f) from gnomAD[25]. To evaluate the accuracy of the AF based on the bioinformatics pipeline used in this study, we counted alternative alleles and estimated AF via visual inspections of read alignment. We found 247 ACs and 55.6% AFs, comparable to the AF estimated by a variant caller (CuteSV[41]; 49.3%).

We also detected a DEL in the gene encoding the drug-metabolizing enzyme cytochrome P450 family 2 subfamily member 6 (*CYP2A6*) (Fig. 6c). The DEL in the *CYP2A6* gene is known to be associated with poor nicotine metabolism[57–59]. The DEL in *CYP2A6* results in a fusion between the 3' UTRs of *CYP2A6* and *CYP2A7*[60], and the alternative allele is referred to as *CYP2A6*4*. This DEL in the *CYP2A6* gene has been reported to be common (15.1–19.0%[58,59], Fig. 6f), but the bioinformatic algorithm used in this study estimated the AF to be only 0.45%. We surmised that this discrepancy is due to an underestimation of AF by the algorithm used because our visual inspection of the *CYP2A6* locus read alignment resulted in an AF concordant with previous estimations (15.5%, Fig. 6f). Nonetheless, we used CuteSV in this study as this variant caller is assumed to be the most accurate available to date.

Focusing on INS, we detected the expansion of triplet repeats in the coding sequence of the Ataxin 3 (*ATXN3*) gene (Fig. 6d). CAG repeat expansion in exon 10 of the *ATXN3* gene is known to cause spinocerebellar ataxia type 3 (SCA3) by resulting in an abnormally long polyQ tract in the encoded protein. Affected individuals are usually heterozygous for the expansion and carry 52–86 CAG trinucleotide repeats in the expanded allele, whereas wild-type (WT) alleles have 12–44 CAG repeats[61]. The numbers of repeats are highly polymorphic[62], and our analysis was not optimized to accurately estimate repeat length; nonetheless, we identified the variation in the repeat as INS with a length of 56 bp (Fig. 6d and Supplementary Fig. 8).

In summary, we constructed an allele-frequency panel focusing on SVs by utilizing the activated T-cell resource in our biobank and nanopore sequencing technology. This strategy was successful in terms of supplying a sufficient amount of high-quality genomic DNA suitable for long-read sequencing analysis and

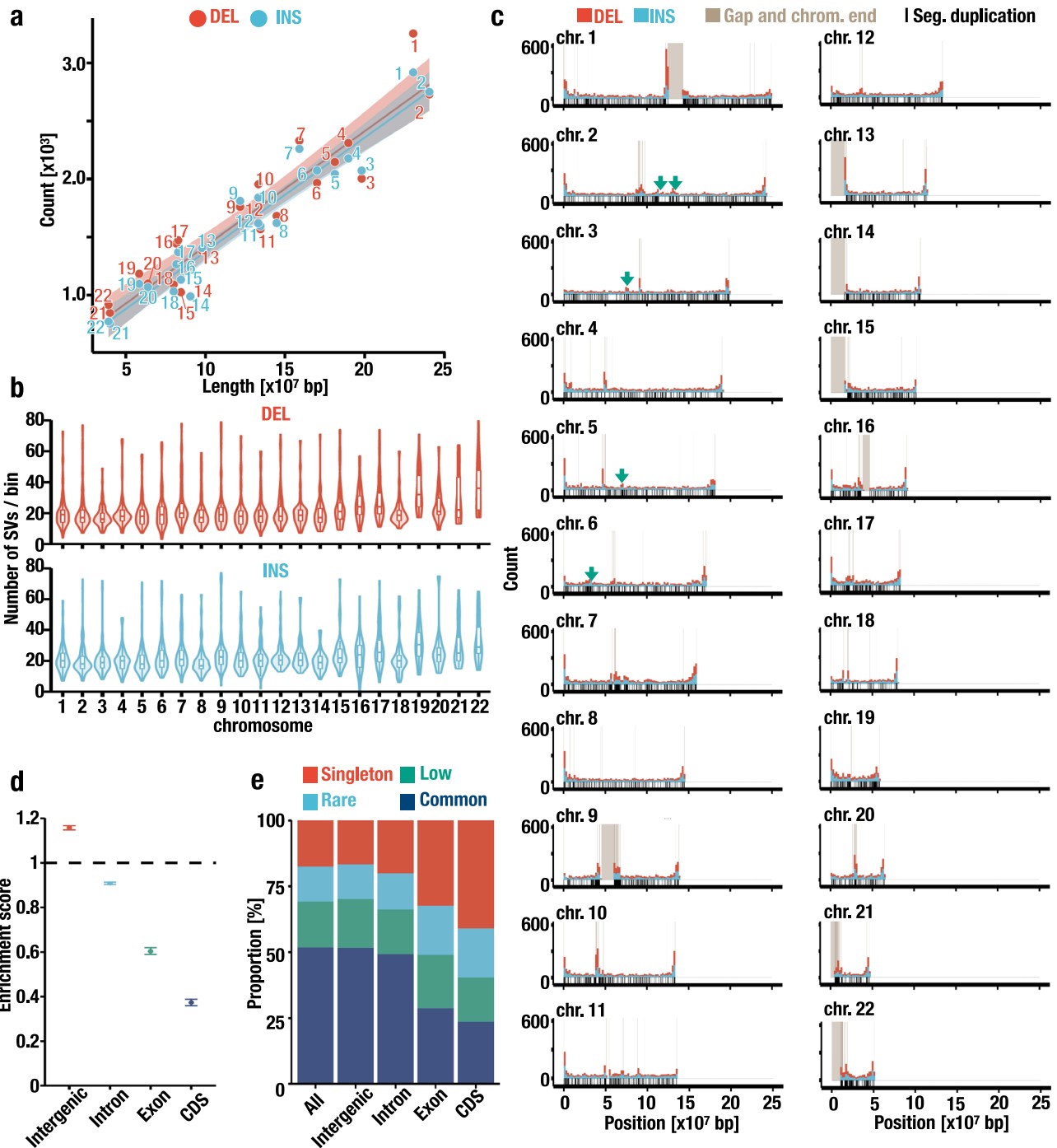

**Fig. 5 Distribution of SVs at the chromosome scale. a** Correlation between the chromosome length and the number of DELs (red) and INSs (blue). The lengths of the N-gaps are excluded from the chromosome length. Regression lines, 95% confidence intervals, and chromosome names are also shown. **b** Number of SVs per 2-Mb bin per chromosome. Each boxplot has a box that represents the interquartile range (IQR) and whiskers that extend 1.5 × IQR from the box edges. The median is shown in a horizontal line in the box. **c** Distribution of SVs at the chromosome scale. The numbers of DELs and INSs per 2-Mb bin are shown in red and blue, respectively. Positions of gaps and chromosome ends in GRCh38 are highlighted in brown. Green arrows indicate SV peaks that are more than 5 Mb away from the gaps. The positions of segmental duplications are shown in black rugs under the histogram. **d** Overlap between SV positions and genomic features. Expected frequencies of SVs that overlapped with each genomic feature (intergenic, intron, exon, and coding sequence [CDS]) were set to 1, and the observed frequencies are shown. The error bars represent the minimum and maximum values of the enrichment score. $n = 1000$. **e** Enrichment of rare SVs in functional genomic features. The proportions of singleton, low, rare, and common SVs that overlapped with genomic features are shown in red, green, blue, and navy, respectively.

high-throughput long-read sequencing at the population scale. We also validated the reliability of the SV panel utilizing trio samples recruited in the TMM biobank to validate the panel by means of Mendelian inheritance error profiling.

## Discussion

Long-read sequencing technology enabled the preparation of an allele-frequency panel focusing on SVs. To pursue long-read sequencing analyses at a population scale, it seems to be desirable

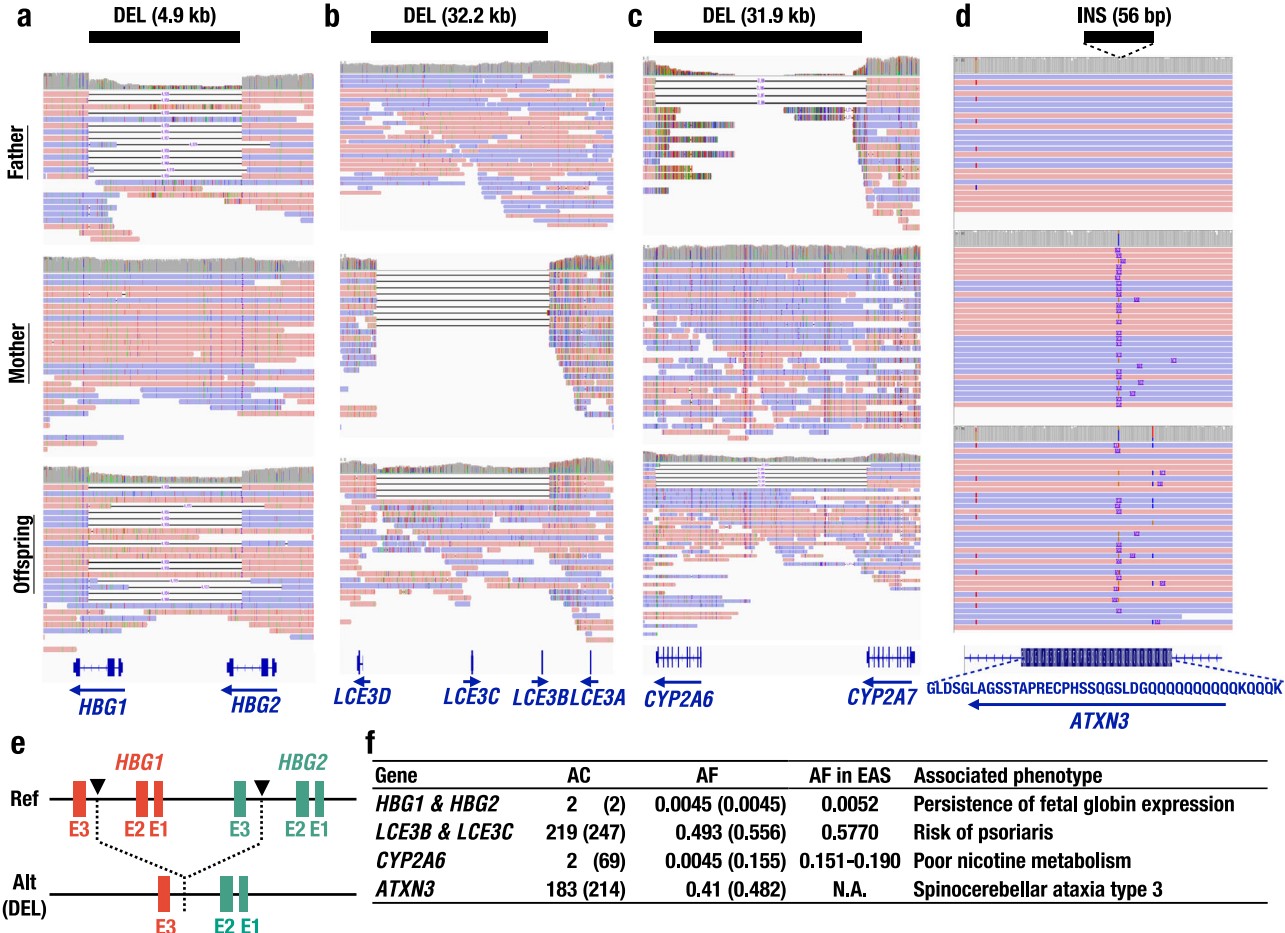

**Fig. 6 SVs associated with clinical phenotypes. a–c** SVs overlapping with the protein-coding sequences of *HBA1* and *HBG2* (**a**), *LCE3B* and *LCE3C* (**b**), *CYP2A6* (**c**), and *ATXN3* (**d**). The positions of the SVs are indicated as black bars, and the sizes are shown in parentheses. **e** Schematic of the 4.9 kb DEL between the *HBG1* and *HBG2* loci. Notably, this DEL produces the *HBG1-HBG2* fusion gene. The breakpoints are shown as arrowheads. **f** Allele frequency and clinical phenotypes associated with the SVs shown in **a–d**. AC allele count, AF allele frequency; and AF in EAS, publicly available AF value in the East Asian population[25, 58, 59]. AC and AF based on manual inspections are shown in parentheses.

to prepare a sufficient amount of high-quality genomic DNA. However, population-scale studies often rely on limited amounts of DNA from peripheral blood. Therefore, we designed an approach that utilizes activated T lymphocytes systematically prepared in the TMM Biobank[14] (Supplementary Fig. 9). We expect that activated T lymphocytes will become a key technology that provides sufficient high-quality DNA samples for long-read sequencing analyses on a population scale. We also exploited trio samples recruited for our BirThree cohort for quality assessment based on MIE profiling. Utilizing this approach, we succeeded in stably producing high-coverage sequencing data with long N50 read lengths on a population scale. We identified mean numbers of 10,923 DELs and 12,133 INSs per individual and succeeded in constructing an allele-frequency panel of Japanese individuals. We validated the allele-frequency panel utilizing trio-based MIE analyses for the careful interpretation of the SVs. We also explored SVs that are likely associated with clinical phenotypes. The SVs identified in this study and their allele frequencies are publicly available on our website, the Japanese Multi Omics Reference Panel (jMorp)[16,63]; the reference panel is referred to as JSV1.

Several reference panels for SNVs and indels have been prepared utilizing short-read WGS technology[1−3], including those of the Japanese population[5,15,63]. However, short-read WGS technology has limitations in the discovery, genotyping, and

characterization of SVs, and it has been difficult to prepare a reference panel for SVs. However, the emergence of long-read WGS technology led us to detect SVs efficiently and enabled the possibility of generating a reference panel for SVs. Indeed, recent long-read WGS studies of Icelandic[35] and Chinese[36] populations have identified more than 20,000 SVs per individual. These detected SVs comprise a greater number than the SVs found per individual by short-read WGS, which has been limited to 4405–7439[21,25]. Therefore, we considered that the application of long-read WGS technology for population-scale genome analyses could support the preparation of a reference panel of SVs, leading us to discover a substantial number of hidden SVs.

To obtain a sufficient amount of high-quality genomic DNA, participant-derived culture cells are an attractive biological resource. Indeed, LCLs have been utilized as a canonical resource in human genome studies[64]. In the present study, we selected activated T cells rather than LCLs because the former can be established much more quickly and with a higher success rate than LCLs. In addition, compared to the complex procedures required for the establishment of LCLs[14], the simple procedure in which only cytokine stimulation is required to introduce proliferation signals into T cells should reduce the probability of technical errors in a cohort-scale analysis. Overall, we predict that currently and in the future, the need for high-quality and large quantities of genomic DNA will increase exponentially with the

advancement of genome analysis technologies. Therefore, the use of activated T cells may become an essential improvement for advanced genome analyses. The activated T-cell resource may contribute to the sustainable development of the biobank, responding to the wide-ranging demands of population-scale genome analyses and avoiding the rapid depletion of biospecimens.

The accumulation of SV datasets at the population scale allowed us to explore genetic differences across ethnic groups. We compared our SV dataset to the dataset published by the deCODE study group[35] and observed that half of the SVs identified in our study were unique to our dataset and that the other SVs overlapped with those in the deCODE dataset. We expected that common SVs would be distributed more commonly across ethnicities than rare SVs, and indeed this was the case. Interestingly, however, a substantial proportion of common SVs were also found to be unique to the TMM dataset. The TMM dataset-specific distribution of common SVs suggests that there exist substantial differences in the occurrence and inheritance of SVs between Japanese and Icelandic populations, implying the presence of marked ethnic diversity in SVs. One caveat is that the bioinformatic pipeline used in each study is different, which should be considered when interpreting this result.

To explore the functional aspects of SVs, the clinical impacts of several SVs identified in this study were examined. We found DELs in the coding regions of genes, including *HBG1/2*, *LCE3B/C*, and *CYP2A6*, and INS in those of *ATXN3*. As the DELs in *LCE3B/C* and *CYP2A6* genes cause large deletions, including in coding regions, functions of these genes must be strongly affected. In contrast, the DEL in the *HBG1-HBG2* locus not only changes the gene structure of the γ-globin gene but affects the gene expression and results in the overexpression of the gene in the adult stage[52,53]. The deletion of the inhibitory element located in the *HBG1* promoter appears to elicit the upregulation of the *HBG1-HBG2* fusion gene in the adult stage, leading to the HPFH phenotype. Supporting this notion, mutations in *HBG1/2* promoters have been shown to upregulate *HBG1* and *HBG2* mRNA expressions[65–67]. These results indicate that SVs disrupting regulatory elements have a functional impact by affecting gene expression profile. Thus, attempts to expand the sample size and carry out follow-up examinations in a prospective cohort study could improve the biological aspects of the SVs discovered in this study.

In conclusion, we constructed an SV database of a Japanese population by utilizing a strategy involving activated T lymphocytes and a trio-based analysis. The expansion of the dataset in the future will improve our understanding of the diversity of the human population and the clinical impact of SVs that affect individual phenotypes, and continuous efforts for further improvement in the bioinformatics pipelines used to analyze long-read sequencing data are anticipated.

## Methods

**Participants**. In total, 333 participants composing 111 trios were recruited through the Birth and Three Generation Cohort (BirThree Cohort)[13], which was led by the Tohoku Medical Megabank Organization (ToMMo) at Tohoku University and Iwate Tohoku Medical Megabank Organization (IMM) at Iwate Medical University. The participants consisted of almost equal numbers of males ($n = 161$) and females ($n = 172$) whose ages ranged from their twenties to eighties (Table 1). Individual written informed consent was obtained prior to enrollment. The protocol was reviewed and approved by the Ethics Committee of Tohoku University Graduate School of Medicine for ToMMo and the Ethics Committee of Iwate Medical University for IMM.

**Establishment and culture of human-activated T cells**. CD19-negative cells were isolated from cryopreserved peripheral blood mononuclear cells (PBMCs) and stimulated with the human T-cell activator CD3/CD28 (Dynabeads, Life Technologies) according to the manufacturer's recommendations. The cells were

**Table 1 Number of individuals analyzed in this study.**

| Age | Male | Female |
|---|---|---|
| 20s | 15 | 23 |
| 30s | 27 | 37 |
| 40s | 8 | 7 |
| 50s | 40 | 44 |
| 60s | 57 | 57 |
| 70s | 13 | 4 |
| 80s | 1 | 0 |
| Total | 161 | 172 |

expanded in a complete RPMI 1640 medium containing 20% heat-inactivated fetal bovine serum (Sigma-Aldrich), 100 U/mL penicillin and 100 μg/mL streptomycin (Nacalai Tesque), 2 mM GlutaMAX I, and MEM containing nonessential amino acids (Thermo Fisher Scientific) and 30 U/mL recombinant IL-2 (PeproTech EC) in a 12-well plate for 3–10 days. After successful cell expansion, the activated T cells were harvested and divided into $5 \times 10^5$ cells per tube for long-term cryopreservation. The frozen cultures were thawed by placing the cryotubes in a water bath at 37 °C and further expanded for the analysis. Subsequently, 1 to $2 \times 10^7$ cells were subjected to genomic DNA extraction.

**Flow cytometry analysis**. The cells were stained with antibodies against human CD3 conjugated with FITC (BD Biosciences, Cat# 555339; 1:50 dilution). The stained cells were analyzed with FACSVerse, and the data analyses were performed using BD FACSuite software (BD Biosciences).

**Nanopore sequencing**. Genomic DNA was extracted from activated T cells using a Gentra Puregene Blood Kit (Qiagen) and sheared using a 29-gauge needle to obtain DNA fragments of the appropriate size. The quality and quantity of the DNA fragments were analyzed using Nanodrop and Qubit fluorometers, respectively, and 2 μg of the DNA fragments were subjected to library preparation using a SQK-LSK109 ligation kit (Oxford Nanopore Technologies [ONT]). Sequencing was conducted using PromethION devices with R9.4.1 flowcells (ONT). Then, the libraries were divided into thirds or fourths and loaded sequentially onto a single flowcell with nuclease flushes[68]. The squiggle data obtained from the PromethION sequencers were subjected to a base-calling step using Guppy software (version 4.2.2) in the "hac" mode.

**Sequencing summaries**. Among a total of 430 runs, we analyzed 323 runs without any problems before and during sequencing (i.e., we excluded runs for the following reason: 23 runs for difficulty during sequencing; 45 runs using flowcells with a low active pore count below the standard of QC; and 39 runs using flowcells with a lot defect due to manufacturing errors). The sequencing summaries were generated using NanoPlot[69] software (version 1.27.0).

**Read alignment and SV calling using nanopore data**. The read alignment and SV calling were conducted following the official pipeline provided by ONT (https://github.com/nanoporetech/pipeline-structural-variation/releases/tag/v2.0.2), with minor modifications. Briefly, the base-called reads with mean quality scores greater than 6 were subjected to downstream analyses after cropping their head and tail 100 bp. The read alignment to the human reference genome (GRCh38) was conducted using LRA[70] (version 2.17-r941) with the option "-ONT". The SVs were called individually using CuteSV[41] software (version 1.0.9) with the "-min_-sv_length 50" option. The individual calls were merged using SURVIVOR software[71] (version 1.0.6) with the option "1000 1 1 -1 -1 -1"; the joint call was conducted using CuteSV software. For the downstream analysis, we focused on autosomes since the SV callers currently in use support only diploid chromosomes. The data were visualized using Integrated Genome Viewer[72] (version 2.4.14) using the options "SAM.HIDE_SMALL_INDEL TRUE", SAM.SMALL_INDEL_-THRESHOLD 10", "SAM.QUICK_CONSENSUS_MODE TRUE", "SAM.-FLAG_LARGE_INDEL TRUE" and "SAM.LARGE_INSERTION_THRESHOLD 10". For the error profiling, deletions, insertions, and mismatches to the reference genome in each mapped sequence were identified using in-house software (https://github.com/informationsea/sequencetoolkit).

**Sample processing**. To eliminate the possibility of sample mix-up and ID mislabeling during the sample processing, including cell culture, we confirmed the collation of SNVs between individual genotypes obtained from the short-read WGS analysis and those obtained from the nanopore sequencing analysis of all samples analyzed in this study[14]. Notably, the short-read WGS analysis and nanopore WGS analysis were conducted independently. The individuals belonging to parents have been previously known to be unrelated individuals based on the mean identity-by-descent (IBD) score[5].

**Mendelian inheritance errors**. The MIEs were quantified using an in-house script (mendelian-check.py) which is provided in Supplementary Software 1. This script counts the number of combinations of parent–offspring trio genotypes for all SVs. We defined the MIEs as the combination of trio genotypes that were inconsistent with Mendelian inheritance. The MIE rates per trio were calculated by dividing the number of MIEs by the total number of combinations of genotypes in the trio.

**Gene enrichment**. This analysis was conducted using an in-house script (gene-enrichment.py) which is provided in Supplementary Software 1. The script calculates the expected numbers of SVs that overlapped with genomic features by randomizing the position of the SVs and counting the number of intervals that overlapped with genomic features, including genes, exons, and CDSs defined by GENCODE[73] (version 36). The enrichment score was defined as the ratio of the numbers of SVs that overlapped with each genomic feature (observed numbers) to the expected numbers.

**Comparison of SVs to published datasets**. The SV datasets published by deCODE[35] and gnomAD[25] were downloaded from the respective websites (https://www.nature.com/articles/s41588-021-00865-4 and https://gnomad.broadinstitute.org). For the comparison with the gnomAD dataset, we constructed an SV dataset based on hs37d5 reference genome in the same way as the GRCh38 version. The datasets subjected to the comparison were merged using SURVIVOR software[71] (version 1.0.6) with the option "1000 1 1 -1 -1 -1", and the SVs with AF > 0 in each dataset were regarded as detected.

**Statistics and reproducibility**. All statistical tests were conducted using R software (ver. 3.5). For correlation tests in Figs. 2f, 3e, f, and Supplementary Figs. 4 and 5, the correlation coefficients and $P$ values were calculated using the test of Pearson's correlation coefficient. For Figs. 3b, 5b, d, the $P$ values were calculated using Holm adjusted Wilcoxon rank-sum test, Kruskal–Wallis rank-sum test, and bootstrap test, respectively. No statistical method was used to predetermine the sample size. No data were excluded. The exact numbers of samples are indicated in individual figure legends and the main text. The definitions of error bars are indicated in individual figure legends.

## Data availability

The SV datasets are available at the jMorp (https://jmorp.megabank.tohoku.ac.jp/202112/) website via a web interface[16,63] and in downloadable variant call format (VCF) files. The VCF files are also available from the Zenodo repository[74] (https://doi.org/10.5281/zenodo.7039938). The data lines in the VCF files contain positions in the genome, reference and alternative alleles, allele frequencies, and the number of families with MIE for each SV site. The sequence data and genotyping results of individuals are under controlled access as they contain information that is restricted by the research participants' consent. The data are available upon request after approval of the Ethical Committee and the Materials and Information Distribution Review Committee of ToMMo. One who wishes to access the data needs to contact jmorp@omics.megabank.tohoku.ac.jp. The approximate response time for accession requests is about two weeks. Numerical data associated with the figures are available in Supplementary Data 3 and the Zenodo repository[75] (https://doi.org/10.5281/zenodo.7049276). An uncropped version of the electrophoresis image in Fig. 1d is presented in Supplementary Fig. 1.

## Code availability

The in-house scripts (mendelian-check.py and gene-enrichment.py) and codes associated with the figures are available in Supplementary Software 1 and 2, respectively. These codes and scripts are also available in the Zenodo repository[75] (https://doi.org/10.5281/zenodo.7049276).

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

## Acknowledgements

We thank Ms. Keiko Tateno and Nanae Osanai for providing technical support. This work was supported by the Tohoku Medical Megabank (TMM) Project from the Ministry of Education, Culture, Sports, Science and Technology (MEXT) and the Japan Agency for Medical Research and Development (AMED; Grant Numbers JP17km0105001 and JP21tm0124005) and by the Advanced Genome Research and Bioinformatics Study to Facilitate Medical Innovation (GRIFIN) project of Platform Program for Promotion of Genome Medicine from AMED (grant number JP16km0405203). All computational resources were provided by the Tohoku University Tohoku Medical Megabank Organization supercomputer system, which is supported by the Facilitation of R&D Platform for AMED Genome Medicine Support conducted by AMED (Grant Number JP16km0405001). This work was also supported in part by JSPS KAKENHI (JP19K16511 and JP22K15376 to A.O.). We appreciate all volunteers who participated in the TMM Project. We also thank the members of TMM for assistance with the study. The member list is available at the following website: https://www.megabank.tohoku.ac.jp/english/a210901/.

## Author contributions

A.O., Y.O., F.K., and M.Y. designed the study. A.O., Y.O., F.K., and M.Y. wrote the manuscript. S.K. organized the TMM BirThree Cohort. K. Kumada and N.M. organized the TMM biobank. N.I. conducted the biobanking of the cell resources. A.O., J.K., K.T., and F.K. conducted the nanopore sequencing analysis. Y.O., S.T., J.T., G.T., and K. Kinoshita implemented the analysis pipelines. A.O., Y.O., S.T., and J.T. analyzed the nanopore sequencing data.

## Competing interests

The authors declare no competing interests.
