## [Transparent Peer Review File · Communications Biology]

Construction of a trio-based structural variation panel utilizing activated T lymphocytes and long-read sequencing technologyResponse to the Reviewers

Response to Reviewer #1

In recent years, long-read sequencing technologies have emerged as a superior method for the detection of structural variants (SVs) within genomes. In this manuscript “Construction of a trio-based structural variation panel utilizing activated lymphocytes and long-read sequencing technology” Otsuki and colleagues performed high coverage (~22X) nanopore sequencing on 111 Japanese trios using activated T lymphocytes as the source of genomic material. They identified a total of 68,571 SVs that are > 50 bp in length (34,341 deletions and 34,230 insertions), with 93.2% of them concordant with Mendelian inheritance. They compared the deletion calls to those from the Iceland population (deCODE dataset), provided a general functional annotation of identified SVs and a few examples of potentially clinically relevant deletions.

We appreciate the reviewer for the thorough review, helpful comments, and evaluation. We addressed all concerns raised by the reviewer. Below, we show the comments by the reviewer in blue, our responses in black, and corrected sentences in the revised manuscript in red. The references are attached at the end of this document.

- 1. One of the suggestions made by the authors for future studies is the use of activated T cells rather than EB virus-transformed B cells. There have been reports suggesting that B and T cell lymphocytes may demonstrate increased levels of genomic instability post VDJ-recombination. The authors should attempt to demonstrate that RAG-mediated or RSS-dependant transposition events during the T cell 'maturation' stages are indeed very infrequent (or even absent) effects in vitro, by directly comparing long-read sequence data generated from donor-matched lymphoblastoid (essentially B-cell) and T-cell samples. The authors are in a good place to ask this question as they are in possession of both the lymphoblastoid and activated T cell material for some of their samples (sequencing a few trios would suffice). This would be very useful both in terms of demonstrating that activated T cells are a superior (and safer) methodology, but also in ensuring that the results are not affected by differences in somatic hypermutation related effects between B and T-cells. Given that a number of large population cohorts (e.g., the 1000 Genomes Project) are lymphoblastoid cell lines, this would be an excellent way to look into potential differences between B- and T-cells and support a shift in the field towards alternative sources of DNA.*

The reviewer notes the possibility that genomic instability in activated T cells may affect the results of this study. We appreciate the comment and worked on a set of new analyses. We would also like to ask the reviewer to understand the advantages of cultured cells, including activated T cells and lymphoblastoid cell lines (LCLs), as fundamental biological resources for genome analyses. These

cell resources are useful in addressing the wide-ranging demands of population-scale genome analyses and avoiding the rapid depletion of biospecimens.

As recommended by the reviewer, we conducted studies to demonstrate that activated T cells, as well as LCLs, are good resources in the context of reliability. To this end, we directly compared nanopore sequence data generated from three pairs of donor-matched LCLs and activated T-cell samples (Supplementary Fig. 2a). Of note, there are donor-matched *de novo* assemblies of all three genomes. The assemblies were constructed from ultradeep sequencing and optical mapping data utilizing genomic DNA from nucleated blood cells and exhibit high contiguity and accuracy (Takayama et al., 2021). We obtained standard SV-call sets from the assemblies and compared them to the SV-call sets from the nanopore sequencing data to calculate the precision and recall score of SV detection. As shown in Supplementary Fig. 2c, activated T cells (blue dots) and LCLs (orange dots) show similar recall and precision scores. Based on this benchmark analysis, we conclude that activated T cells are an acceptable resource for genome analyses like LCLs.

We described the results in the revised manuscript, including the benchmark analysis of the SV calling algorithms for the response to Comment #3.

Line 154: To address whether the utilization of activated T cells is a good methodology for genome analyses, we designed a benchmark analysis using three independent sets of genomic DNA samples obtained from activated T cells and LCLs (Supplementary Fig. 2a). Notably, there are donor-matched and high-quality *de novo* assemblies for all three genomes (Takayama et al., 2021). In this benchmark analysis, we obtained standard SV-call sets from the assemblies and compared them to SV-call sets from nanopore sequencing data to calculate the precision and recall scores of the SV detection (Supplementary Fig. 2a). To select SV call pipeline, we experimentally compared the efficiency and accuracy of CuteSV (Jiang et al., 2020) with Sniffles algorithms (Sedlazeck et al., 2017), both of which are widely used algorithms available for the nanopore sequencing data. As shown in Supplementary Fig. 2b, CuteSV reproducibly showed higher recall and precision scores than Sniffles in the detection of DELs. This result is concordant to the previous benchmark studies (Bolognini & Magi, 2021; Jiang et al., 2020), so that we decided to utilize CuteSV algorithm. Utilizing this algorithm, we next compared the precision and recall scores of activated T cells and those of LCLs. As shown in Supplementary Fig. 2c, we observed that activated T cells and LCLs showed almost similar recall and precision scores. Therefore, we concluded that activated T cells were an acceptable resource for a genome analysis like LCLs.

2. *Comprehensive and accurate identification of SV from long-read data remains a challenge. The authors focused on the detection of only deletions and insertions but not other types of*

SVs (e.g., inversions) nor did they explain why other SVs were not included in the analyses. With the generated data they could attempt to identify at least a proportion of those. Then, for downstream analyses (e.g., the comparison to the deCODE dataset, and the Results section “SVs associated with clinical phenotypes”), the authors focused on deletions and ignored insertions. Including insertions and other types of SVs in the downstream analyses would be important and substantially strengthen the manuscript.

We agree with these comments. As noted by the reviewer, in this study, we focused on deletions and insertions but not on other types of SVs because the callers we used could not identify inversions and duplications; however, the callers showed high performance in calling insertions and deletions (please see reply to Comment #3). Therefore, we would like to work on inversions and duplications in our next study.

Following the advice by the reviewer, we worked extensively to address the second comment. We included insertions in the downstream analyses, including the comparison to the deCODE dataset and the search for SVs associated with clinical phenotypes.

In the comparison with the deCODE dataset, we are happy to report our new results to the reviewer. Concordant results were obtained between insertion and deletion. We observed that more than half of the SVs detected in the TMM dataset, which includes both INS and DEL, overlapped with those in the deCODE dataset (Fig. 3c). We also compared the MAFs of the unique DELs and INSs to those of overlapping ones and found that the SVs in the common category were shared preferentially with those in the deCODE dataset, while in contrast, those with a low MAF and those in the rare and singleton categories tended to be unique in the TMM dataset (Fig 3d). As this comment is heavily expanded in Comment #6, we provide further details in the response to Comment #6.

Line 255: As shown in Figure 3c, of all SVs in our dataset (INS and DEL; shown as “TMM”), 38,304 (53.5%) were also found in the deCODE dataset, while 33,279 were unique to the TMM dataset. Next, we compared the MAFs of the unique DELs and INSs to those of overlapping ones. The results revealed that the SVs in the common category were shared preferentially with those in the deCODE dataset; in contrast, those with a low MAF and those in the rare and singleton categories tended to be unique in the TMM dataset (Fig 3d). Thus, the comparison of the deCODE and TMM datasets revealed significant differences in the ethnic distribution of SVs, even though high-MAF SVs are shared relatively widely across ethnicities.

Regarding the search for SVs associated with clinical phenotypes (b), we added the insertion

of triplet repeats in the coding sequence of Ataxin 3 (*ATXN3*) gene. While the expansion of the CAG repeat in the *ATXN3* gene is known to cause spinocerebellar ataxia and is highly polymorphic in the general population (Mitsuhashi, Frith, & Matsumoto, 2021), we detected the repeat expansion as an INS with a length of 56 bp. (Fig. 6d). The manuscript was revised as follows:

Line 393: Focusing on INS, we detected the expansion of triplet repeats in the coding sequence of the Ataxin 3 (*ATXN3*) gene (Fig. 6d). CAG repeat expansion in exon 10 of the *ATXN3* gene is known to cause spinocerebellar ataxia type 3 (SCA3) by resulting in an abnormally long polyQ tract in the encoded protein. Affected individuals are usually heterozygous for the expansion and carry 52–86 CAG trinucleotide repeats in the expanded allele, whereas wild-type (WT) alleles have 12–44 CAG repeats (Kourkouta et al., 2019). The numbers of repeats are highly polymorphic (Mitsuhashi et al., 2021) and our analysis was not optimized to accurately estimate the repeat length; nonetheless, we detected the variation in the repeat as INS with a length of 56 bp (Fig. 6d and Supplementary Fig. 8).

3. *For the deletion and insertion detection, the authors only used one algorithm CuteSV and claimed its superiority (lines 317-318: “Nonetheless, we used CuteSV in this study, as this variant caller is assumed to be the most accurate available to date.”). Such a statement should be supported by references. Best practices in the field incorporate multiple SV calling algorithms. Have the authors compared the CuteSV results with other widely-used SV calling algorithms, such as pbsv, sniffles and SVIM etc? Have they evaluated the performance of one algorithm vs combining multiple algorithms? The authors did not justify why they used only one algorithm and ideally this new method should be compared to existing methods to demonstrate its efficiency and accuracy, while the authors have not presented any such comparative analysis.*

We thank the reviewer for these professional comments. The central point of this comment is the lack of a description of why we used only one algorithm (CuteSV) to call SVs and the lack of the demonstration of the efficiency and accuracy. While there are multiple SV-calling algorithms as the reviewer mentioned we would like to emphasize that our allele frequency panel aims to be utilized for the diagnosis of rare diseases in clinical sequencing. Therefore, the use of a simple bioinformatics pipeline has an advantage. Therefore, we selected one algorithm for the analysis based on several benchmark studies of SV calling pipelines (Bolognini & Magi, 2021; Jiang et al., 2020).

In this revision, to strengthen the rationale for this decision, we conducted further benchmark analyses of CuteSV along with the widely used Sniffles algorithms using the three

individual nanopore sequencing datasets in the settings mentioned above (please see the response to Comment #1). As shown in Supplementary Fig. 2b, CuteSV reproducibly showed higher recall and precision scores than Sniffles, especially for DEL. These results further support our selection of CuteSV as an SV-calling algorithm. We described this benchmark comparison study in the revised manuscript as follows:

Line 160: To select SV call pipeline, we experimentally compared the efficiency and accuracy of CuteSV (Jiang et al., 2020) with Sniffles algorithms (Sedlazeck et al., 2017), both of which are widely used algorithms available for the nanopore sequencing data. As shown in Supplementary Fig. 2b, CuteSV reproducibly showed higher recall and precision scores than Sniffles in the detection of DELs. This result is concordant to the previous benchmark studies (Bolognini & Magi, 2021; Jiang et al., 2020), so that we decided to utilize CuteSV algorithm.

- I wonder why the authors are still using the old hs37d5 (released in 2009) for their analysis when the human reference GRCh38 version has been widely used in most human genome studies since its release in 2013. The authors should re-analyse all the data using a more recent human reference (at some point even switching to T2T-CHM13 reference <https://doi.org/10.1101/2021.05.26.445798>). Additionally, if the authors mapped the sequence data to the GRCh38 reference, the results would be readily comparable to other published studies in the field without the need of liftover datasets from GRCh38 to hg19 or vice versa.*

We agree with this comment and changed the reference genome to GRCh38. Accordingly, we reanalyzed all data in the revised version.

Regarding the characterization of SVs in the Japanese population (Fig. 2d, 2e, 3a, and 3b), we found that the numbers of SVs, size distribution, and distribution of allele frequencies in the GRCh38 version were almost comparable to those in the hs37d5 version. Nonetheless, there were small changes in the numbers of SVs; thus, we revised the related sentences in the main text (please see the changes in the main text sentences starting from **Lines 201, 207, and 239**). In addition, we included a reanalysis using GRCh38 in the comparison between the TMM and deCODE datasets (Fig. 3c and 3d).

Regarding the Mendelian inheritance error (MIE) analysis (Fig. 4), distribution and functional annotation of SVs (Fig. 5), and clinical implication of SVs (Fig. 6), we also obtained concordant results between the GRCh38 version and hs37d5 version, and we applied minor revisions in the related sentences starting from **Lines 290, 324, and 342**.

- 5. There is a substantial difference in the overall proportion of deletion and insertion variants identified in this study compared to previous studies (e.g., deCODE - <https://doi.org/10.1038/s41588-021-00865-4>, Ebert et al 2021 - DOI: 10.1126/science.abf7117). The current study identified a total of 68,571 deletion and insertion variants, with roughly 50-50 ratio between the categories, while this ratio is closer to 40-60 in both abovementioned studies. Considering the total number of identified SVs, this is a substantial difference that would be important to elaborate on. As also mentioned by the authors, calling insertions accurately is more challenging than calling deletions, so perhaps a substantial number of insertion variants have remained unidentified in the current study indicating technical limitations (see the previous comments about SV calling algorithms and the use of hs37d5 as reference).*

The reviewer noted a contradiction in the ratio of identified DELs and INSs between this study and previous studies in which a bias toward INSs was caused by missing or incomplete sequences residing in the international reference genome. The reviewer surmised that in our dataset, a substantial number of INSs were missing due to the use of hs37d5 or the bias in the SV calling algorithm.

To address these possibilities, we evaluated the numbers of DELs and INSs in our GRCh38 version but again observed balanced numbers (37,981 and 36,220, respectively) as was the case using hs37d5. We also addressed the issue of SV calling algorithms and evaluated the benchmark analysis shown in the response to Comment #2. As shown in Supplementary Fig 2b, we observed a slightly lower recall score of DEL compared to that of INS using the CuteSV algorithm, indicating that a part of INS may be missing in our pipeline. Considering these points, we revised the sentence as follows:

Line 208: Then, we merged these SVs of the 333 individuals into a nonredundant set of SVs to produce a variant repository composed of 37,981 DELs and 36,220 INSs, showing a balanced number of DELs and INSs (Fig. 2e). In this regard, several studies have identified more INSs than DELs (Beyter et al., 2021; Ebert et al., 2021). A plausible explanation for this discrepancy may be the lower recall scores in INS detection than DEL detection in our study (Supplementary Fig. 2b). We surmise that biases in SV calling remain and expect the development of elaborate bioinformatics algorithms.

- 6. The authors have only compared their deletion calls to the deCODE dataset but not the insertion calls and no comparison has been performed with other recent publications or SV databases (e.g., DGV, gnomAD). This would be a very important addition.*

The first part of this comment is related to Comment #2. We agree with this comment and carried out analyses that includes insertions. In the analyses, we obtained concordant results in terms of INs to those obtained in the analysis of DELs (Fig. 3c and 3d; please see the response to Comment #2). Additionally, we compared the allele frequencies in the deCODE dataset with those in our TMM dataset utilizing common SVs ($MAF \geq 0.05$). As a result, we found only a weak or moderate correlation between these two datasets (Fig. 3e), suggesting that this discrepancy may be due to ethnic differences in allele frequencies (AFs).

The reviewer further advised to examine the ethnic difference in the AFs by using the gnomAD-SV dataset. Accordingly, we carried out a comparison of our dataset and the gnomAD-SV dataset. Notably, the samples in the gnomAD-SV dataset were derived mainly from European and African/African American samples, while less than 10% of the samples were from East Asian samples. Therefore, in the comparison to the whole gnomAD-SV dataset, only modest correlations between the TMM dataset and the whole gnomAD-SV dataset were observed (Supplementary Fig. 5), which is concordant with the observation in the comparison between the TMM dataset and deCODE dataset. However, if we compare the TMM dataset to the East Asian subset in the gnomAD-SV dataset, much higher correlations were observed between these two datasets (Fig. 3f). This observation indicates that the AFs in the TMM dataset nicely reflect the ethnicity of AFs in the Japanese population. We revised the manuscript as follows:

Line 264: For further analysis focusing on the ethnic diversity of SVs, we extracted common SVs ($MAF \geq 0.05$) and compared the allele frequencies (AFs) between the deCODE dataset and the TMM dataset (Fig. 3e). We observed only modest correlations between these two datasets (Pearson correlation coefficient [cor] = 0.50 and 0.40 for DEL and INS, respectively), indicating that the common SVs showed substantial differences in AFs between the ethnicities.

Next, we compared the TMM dataset to that of gnomAD-SV (Collins et al., 2020), which was based on short-read WGS of more than 14,000 samples. Notably, the samples are mainly derived from European and African/African-American samples, while less than 10% of the samples are derived from East Asian samples (EAS). In the comparison between the TMM dataset and the whole gnomAD-SV, only modest correlations ($cor = 0.61$ and 0.57 for DELs and INs, respectively) were observed (Supplementary Fig. 5), showing very good agreement with the observation in the comparison to the deCODE dataset. In contrast, we observed much higher correlations ($cor = 0.74$ and 0.78 for DELs and INs, respectively) in the comparison of the TMM dataset and the East Asian subset in the gnomAD-SV

(Fig. 3f), demonstrating that the AFs in the TMM dataset reflect ethnicities in AFs.

7. *Data availability: it is not clear whether the generated SV dataset is publicly available or not. At the Discussion section, lines 345-347, it says “The SVs identified in this study and their allele frequencies are publicly available at our website, the jMorp...”. In the Data availability section, lines 491-495, it says “The SV dataset will be available at the jMorp”. I was not able to find the SV dataset on the JMorp website. Can the authors make sure to submit the raw data and/or SV dataset before submission so that reviewers can assess the data and that the data will be available to the general scientific community after acceptance of the publication?*

We fully agree with the reviewer. We uploaded both the hs37d5 and GRCh38 versions of the SV datasets to our jMorp website (<https://jmorp.megabank.tohoku.ac.jp/202112/>), and the SV data can be visualized through our web-based interface and downloaded. We also revised the sentence referring to the data availability (**Line 596**).

8. *The authors have not explained why the sex chromosomes have been excluded from their analyses.*

We thank the reviewer for this professional comment. While we recognize the importance of including sex chromosomes, to the best of our knowledge, the SV callers currently in use, such as CuteSV, Sniffles, and LRcaller, only support diploid chromosomes. These SV callers do not support the genotyping of SVs on the sex chromosomes. For chromosome X, we only have 172 female samples comprised of mother-child pairs, rendering the accurate estimation of AFs difficult due to insufficient statistical power. Therefore, in this paper, we decided to focus on autosomes. We would prefer to leave analyses including the sex chromosomes for future studies. We succinctly mentioned this situation in the text (**Line 555**).

9. *The Results section “SVs associated with clinical phenotypes” could be improved by including examples of insertion variants and more detailed analysis of associations to phenotypes. Is there any phenotypic information available for the sequenced individuals that could be used to investigate associations with SVs? Admittedly, the sample size is small, but nevertheless, some interesting examples could be provided.*

We thank the reviewer for this suggestion. To provide an example of an insertion variant, we added the insertion of triplet repeats in the *ATXN3* gene, which is related to spinocerebellar ataxia.

Unfortunately, it was difficult to address the association between SVs and disease phenotypes since this study was designed based on prospective cohorts of general populations. This study focuses on the characterization of SVs in the general (healthy) Japanese population, and the sample size with long-read sequencing data is relatively small. Nonetheless, we agree with the advice and aim to address this topic in future research, and we described this information in the Discussion.

Line 490: Thus, attempts to expand the sample size and carry out follow-up examinations in a prospective cohort study could improve the biological aspects of the SVs discovered in this study.

10. *Lines 265-267 “This observation supports the hypothesis that there are structure-variant hotspots in the human genome and that these peaks may correspond to these hotspots. In fact, the blue arrow position on chromosome 6 involves human leukocyte antigen (HLA) loci.” – did they look at the other “hotspot” regions in more detail? If not, then perhaps it would be interesting to do so. According to Figure 5b, there are only 4 of such hotspots, so a more detailed analysis of these “hotspot” regions would be doable, and would definitely add value to the paper.*

We thank the reviewer for this professional comment. We identified five hotspot regions, including HLA loci, which showed the accumulation of SVs despite being distantly located from the gap and chromosome ends (Fig. 5c in the revised version, green arrows). To search for genomic features in these regions, in this revision, we evaluated the relationship between the positions of the peaks and those of segmental duplications (SDs). As shown by the rug plot indicating the position of SDs under the histogram (Fig. 5c), we found that the positions of the five hotspots correspond to those of SDs. This finding is consistent with the knowledge that SVs tend to occur in not only chromosome ends and sites adjacent to the gaps but also regions enriched with SDs (Audano et al., 2019). The regions enriched with SDs are intractable with current long-read sequencing technology. We surmise that the difficulties in SV detection in these regions might lead to the overestimation of SVs. We revised the manuscript as follows:

Line 334: Nonetheless, we found five peaks of SVs located in positions far from the gaps and chromosome ends (Fig. 5c, green arrows). The green arrow position on chromosome 6 involves human leukocyte antigen (HLA) loci. Regarding the position, a closer analysis of our long-read sequence data revealed that these five

peaks are located in regions that harbor high-level segmental duplications (Fig. 5c). The regions with SD accumulations are intractable with the current long-read sequencing technology. We surmise that the difficulty in SV detection in these regions might result in the overestimation of SVs.

11. It would be very interesting to add a more thorough analysis of SVs showing Mendelian inheritance errors (MIE) – what kind of SVs are they, the sequence composition, genomic location, how many are located in coding sequences, is it possible to separate biological from technical errors etc.

We thank the reviewer for this thoughtful advice. To address this comment, we evaluated the incidence of the MIEs of each SV by calculating the error family ratio (number of trios with MIE/total number of trios analyzed) and carried out characterizations of the SVs showing MIEs. First, regarding the SV types, we observed concordant values between DELs and INSs (Supplementary Fig. 6a). Second, to address the genomic distribution of SVs with MIEs, we plotted the error family ratios precisely on genomic locations. As shown in Supplementary Fig. 6b, we identified SVs located near gaps and chromosome ends that frequently have MIEs. The SVs in the regions near gaps and chromosome ends were often called based on low coverage sequencing reads (Supplementary Fig. 7a and b). Therefore, we surmise that the high incidences of MIEs were derived, at least partly, from erroneous SV calls due to the difficulty in read mapping. The manuscript was revised as follows:

Line 300: We attempted to examine the characteristics and genomic locations of the SVs that showed MIEs. Taking advantage of the trio analysis, we evaluated the incidence of MIEs in each SV by calculating the error family ratio (number of trios with MIE/total number of trios analyzed). For the SV types that showed MIEs, we observed concordant values in the DELs and INSs (DEL, $3.2 \pm 5.3\%$; INS, $3.7 \pm 5.2\%$ of trios showed MIEs; Supplementary Fig. 6a). To address the genomic distribution of SVs with MIEs, we precisely plotted the error family ratios on genomic locations (Supplementary Fig. 6b). We identified SVs located near gaps and chromosome ends that frequently accompany MIEs. We also found that the SVs in the regions near gaps and chromosome ends were often called based on low coverage sequencing reads (Supplementary Fig. 7a and b). Therefore, we surmise that the high incidences of MIEs were derived, at least partly, from erroneous SV calls due to the difficulty in read mapping.

12. Why did the authors estimate allele frequencies for 65,383 SVs, and not all 68,571 SVs?

We apologize for the confusion. To avoid the overestimation of AFs by double counting the variants shared in a parent-offspring pair, we did not use all SVs to estimate the AFs. This strategy is generally applied for the construction of an allele frequency panel, including in our previous work (Tadaka et al., 2019). To clarify this point, we revised the sentence as follows:

Line 234: To avoid double counting the SVs shared between parents and offspring and, thus, prevent the overestimation of the allele frequencies of the SVs in the general population, we extracted SVs observed in 222 unrelated individuals (*i.e.*, fathers and mothers) from the repository to evaluate MAF.

13. *It is not clear how the authors overcome the high error rate of nanopore reads. In line 160, they mentioned that they used a Phred score of 6 to filter reads but no further analyses were performed.*

The reviewer noted a way to overcome the high error rate of nanopore reads other than filtering the reads. Indeed, we recognize that there are several error-correction tools. For instance, Ratatosk (Holley et al., 2021) and FMLRC (Wang, Holt, McMillan, & Jones, 2018) have been developed to improve the accuracy of SNP/Indel detection. Since we focus on SV and SV-calling algorithms, such as CuteSV, which was developed to address erroneous nanopore reads, we did not apply such tools in this study. In addition, in the response to Comment #3, we envisage that a simple bioinformatics pipeline may have an advantage in the context of clinical sequencing.

14. *Line 163: the authors did not explain how well nanopore reads cover unique and repetitive regions (which are usually very difficult to assemble).*

The reviewer noted the lack of an explanation of the performance of nanopore reads in terms of the alignment of repetitive sequences. We mentioned such an advantage of nanopore reads in the Introduction, but we did not verify the statement directly in this study.

Therefore, we softened the description as follows:

Line 82: These long-read sequencing technologies are able to produce an average read length of several thousand base pairs or greater, which are more likely to span the break points of SVs with high-confidence alignments, helping in capturing larger SVs better than short reads alone (Audano et al., 2019; Chaisson et al., 2019; Mahmoud et al., 2019; Wu et al., 2021).

15. *Line 204: “Because comparison of INs is technically challenging,”. The authors should elaborate why this is the case.*

We performed a comparison of our INS dataset and the deCODE dataset. We obtained concordant results between the DELs and INs (as described in the response to Comments #2 and #6). Please see the paragraph starting from **Line 252** in the revised manuscript.

16. Lines 254-276: the authors could provide more details in this part, and could use a different plot to show the number of SVs on each chromosome to highlight how some chromosomes have more SVs compared to others even when the size of the chromosome is smaller.

The reviewer requests a detailed description of the distribution of SVs on a chromosome scale, including the number of SVs on each chromosome. Regarding the distribution of SVs on a chromosome scale, we carried out further analyses of the increase in SVs in gap proximity by estimating the sequencing coverage. As shown in Supplementary Fig. 7, the numbers of reads supporting a variant call are low in these regions, indicating the possibility that difficulty in read mapping can affect the detection of SVs. We also added an evaluation of the SV distribution on each chromosome to address the comment. As shown in Fig. 5b, we observed a differential distribution of SVs on each chromosome. Based on this new information, we revised the manuscript as follows:

Line 324: Although the number of SVs correlated with the chromosome length (Fig. 5a), **the SVs were differentially distributed on each chromosome (Fig. 5b; $P < 2.2 \times 10^{-16}$ for DELs and $P = 6.87 \times 10^{-16}$ for INs; Kruskal-Wallis rank-sum test).**

We also observed several peaks, indicating the high density of SVs on chromosome ends and sites adjacent to the gaps remaining in the reference genome, and **this observation is concordant with the observation in a previous study (Audano et al., 2019) (Fig. 5c, shown).** To evaluate the nonrandom distribution of the SVs in detail, we plotted the number of sequencing reads supporting the variant call of each SV (Supplementary Fig. 7a and 7b). The sequencing depths were decreased in the genomic regions adjacent to gap and chromosome ends, suggesting that the difficulty in read mapping might result in **the erroneous detection of SVs in these regions.**

17. Lines 272-274 “Thus, SVs located in intergenic regions were overrepresented, and SVs in introns, exons, and CDSs were underrepresented (Fig. 5d). We also observed elevated rates of rare alleles in exons and CDSs (Fig. 5e).”. Surely this is not unexpected and perhaps is worth comparing to the deCode or other published studies.

We obtained a result consistent with the deCODE analysis. Nonetheless, it is difficult to compare results in detail due to differences in the analysis conditions, including genome and gene/transcript annotations, between the two datasets. We revised the sentences by citing the deCODE study as follows:

Line 341: Next, we examined the localization of these SVs within intergenic regions, introns, exons, and protein-coding sequences (CDSs). *Of 70,722 SVs, 32,964 (46.6%) were in intergenic regions, while 39,475 (55.8%), 5,413 (7.7%), and 1,823 (2.6%) SVs overlapped with introns, exons, and CDSs, respectively ($P < 0.002$, bootstrap test), which is concordant with a previous study (Beyter et al., 2021).*

Minor comments:

18. *Lines 70-72 - the statement requires a reference.*

We added a reference (Conrad et al., 2006, Nat. Genet.) that supports the statement (**Line 73**).

19. *Line 157 - the authors could provide the exact number of males and females*

We mentioned the exact numbers of males and females in Table 1. We added these numbers to the main text (**Line 501**).

20. *Lines 190-195 - please add the size for different SVs.*

We added the size of the SVs in each category (**Line 245**).

21. *Lines 266 - typo, "structure-variant" is not a term.*

We deleted this term.

22. *Line 270-271 "Of 68,571 SVs, 28,720 are located in intergenic regions; 38,524, 3,099, and 825 SVs overlap with introns, exons, and CDSs, respectively." Please add percentages to this sentence.*

We revised the number of SVs based on the GRCh38 version analysis and added percentages (**Line 341**).

23. *Line 283 - Please add the Table # for the supplementary Table.*

We added numbers for the Supplementary Tables (**Lines 179 and 355**).

24. Line 443 – the degree symbol has been lost.

We corrected the degree symbol (**Line 521**).

25. Line 588 and line 596 - reference 32 and 36 are duplicates.

We retained the latter (Beyter, D. *et al. Nat Genet* (2021)).

26. Figure 2b and 2c – I would suggest adding the means to the plots as vertical lines as well.

We added vertical lines that indicate the “mean” in Figs. 2b and 2c.

27. Figure 4b – is it minimum coverage that is plotted on the x-axis, or is it mean coverage?

We plotted the minimum coverages of the trios on the x-axis in Fig. 4b. We changed the x-axis label to enhance clarity.

28. There is no Figure 6f though it is referred to at least twice (line 299 and line 305).

We apologize for the oversight and corrected the numbers.

29. Figure 7 – feels unnecessary – I would suggest to either move it to supplementary or remove entirely.

We moved Fig. 7 describing the overall design of the study to Supplementary Fig. 9.

30. Supplementary Figure 1 – the colouring of deletions and insertions is potentially misleading and leaves an impression like in some regions only one or the other SV type is found. Unless this is the case I would suggest using a partially transparent colouring.

We thank the reviewer for this professional comment. We intended to show the genomic regions located near the gaps and ends of chromosomes (Supplementary Fig. 1 in the old version) and that the numbers of SVs increased in these regions (Fig. 5c in the old version). Since we noticed that these illustrations were difficult to understand, we included this information in Fig. 5c in the revised version as brown ribbons and removed Supplementary Fig. 1 and Fig. 5c in the old version. We

also revised the sentence as follows:

Line 327: We also observed several peaks, indicating the high density of SVs on chromosome ends and sites adjacent to the gaps remaining in the reference genome, and **this observation is concordant with the observation in a previous study (Audano et al., 2019) (Fig. 5c).**

Line 334: Nonetheless, we found five peaks of SVs located in positions far from the gaps and chromosome ends (Fig. 5c, **green** arrows).

31. *Supplementary Table. Protein coding genes overlapped with SVs – it would be useful to add the genomic location of these genes, size of identified SVs and the allele frequency/number of observations of overlapping SVs.*

We agree with the reviewer and added related information (the genomic location of the genes and the SVs, and size and allele frequency of the overlapping SVs) to Supplementary Table 2.

Response to Reviewer #2

The authors present a study based on the Oxford Nanopore Technologies (ONT) long-read sequencing of 333 Japanese individuals from 111 parent-offspring trios. They designed a sample preparation protocol that enables high quality and high throughput DNA acquisition and were able to use the recommended ONT SV detection pipeline with good SV detection accuracy, shown by the Mendelian error rate estimates and genotype distribution. The results shown confirm the findings of previous cited work. While this work is a first example of a large scale long-read sequencing for the Japanese population, it should focus on what further knowledge is gained from such a Japanese reference SV dataset from long-reads, combined with utilizing activated T lymphocytes.

We appreciate the reviewer for the thorough review, helpful comments, and evaluation. We addressed all concerns raised by the reviewer. Below, the reviewer's comments are shown in blue, our responses are shown in black, and corrected sentences in the revised manuscript are shown in red. The references are attached at the end of this document.

For this manuscript as is, Communications Biology is more suitable among the suggested ones. (i.e. Nature Genetics, Nature Communications, and Communications Biology). Although the data analysis results are currently confirmatory of previous long-read studies, if focus on a Japanese SV reference dataset is made, and what further knowledge can be obtained from its analysis it could increase the contribution of this study in the field. Otherwise, if the authors prefer to highlight the utilization of activated T lymphocytes for long-read sequencing on SV discovery, then the paper can benefit from focusing on sample preparation.

We appreciate the reviewer for the kind evaluation of the value of this work. We would like to inform the reviewer that we have already opened the JSV1 reference panel in the jMorp database (<https://jmorp.megabank.tohoku.ac.jp/202112/>) based on the current long-read sequence data. We also identified several phenotypes related to INs and DELs in Japanese individuals. We believe that the establishment of a unique DNA preparation approach by utilizing activated T-lymphocytes for cohort size long-read sequence analyses is one of the salient contributions of this work as the method prevents the depletion of DNA samples, which is essential for biobank-based genome studies.

The high-quality DNA derived from activated T cells enabled us to perform high-quality long-read sequencing. As shown in Figure 2b, our strategy led to relatively long (read N50 of 25.8 ± 3.9 kb) sequence reads compared to previous works, demonstrating that the use of activated T-cell resources enables the stable supply of high-quality DNA suitable for SV analyses at the population scale.

To ascertain the benefit of high-quality DNA in SV analyses, we assessed the correlation between the read length and SV detection ability. As shown in Fig. 2f, we observed a strong correlation between the

read N50 and the mean size of INSSs and a moderate correlation between the read N50 and the mean size of DELs. These results suggest that SV analyses using longer reads have an advantage in the detection of large SVs compared with those using shorter reads. To further verify this finding, we also evaluated the correlation between the read length and SV detection ability by detecting large SVs ranging from 5.9 to 6.1 kb and small SVs from 280 to 350 bp (Supplementary Fig. 4a and b, respectively). The SVs belonging to the former fraction contain LINE-1-related SVs, and the latter fraction contains Alu-related SVs. As a result, we found that the number of large INSSs was correlated with the read N50, whereas that of DELs and small INSSs was not correlated. These results indicate that SV analyses using longer reads utilizing activated T cells are beneficial for the comprehensive detection of large SVs, especially in the case of INSSs. We added the following sentences to describe these results in the revised manuscript:

Line 218: To ascertain the benefit of using high-quality DNA in SV analyses, we examined the correlation between the read length and SV detection ability. As shown in Fig. 2f, we observed a strong correlation between the read N50 and the mean size of the INSSs and a moderate correlation between the read N50 and the mean size of the DELs (Pearson correlation coefficient [cor] = 0.54 and 0.80 for DEL and INS, respectively). These results suggest that an SV analysis using longer reads has an advantage in the detection of large SVs compared with that using shorter reads. To further verify this finding, we also evaluated the correlation between the read length and SV detection ability by detecting large SVs ranging from 5.9 to 6.1 kb and small SVs ranging from 280 to 350 bp (Supplementary Fig. 4a and b, respectively). The SVs belonging to the former fraction contain LINE-1-related SVs, and those belonging to the latter fraction contain Alu-related SVs. We found that the number of large INSSs was correlated with the read N50 (cor = 0.54), whereas that of the DELs (cor = 0.18 and 0.08 for large and small DELs, respectively) and small INSSs (cor = 0.25) was not correlated with the read N50. These results indicate that an SV analysis using longer reads utilizing activated T cells is beneficial for the comprehensive detection of large SVs, especially in the case of INSSs.

1. *l. 26: enable better* characterization of SVs. (characterization of SVs have already been achieved to a degree using short read sequencing.)*

We reworded the sentence as mentioned (**Line 27**).

2. *l. 39-40: This sentence is unclear. Also, previous population scale short read and long read sequencing studies analyzing SVs already achieved that. The authors can focus what is achieved in this study on top of previously published literature.*

To clarify the aim and achievement of this study, we revised the sentence as follows:

Line 38: Our data provide a catalog of SVs in the general Japanese population, and the present approach **using the activated T-lymphocyte resource will contribute to biobank-based** human genetic studies focusing on SVs at the population scale.

3. l. 43: reword "genome medicine". Perhaps "genomic medicine"?

We reworded the sentence as mentioned (**Line 44**).

4. l. 46: rewrite: "genetic diversity and disease biology".

We rewrote the sentence as mentioned (**Line 47**).

5. l. 51: I do not understand why filtering neutral variants is listed in this sentence. Neutral variants, i.e., variants that are neither beneficial nor detrimental, are not filtered out from a generic catalog of variants.

Since the establishment of TMM biobank, a biobank collecting samples and information of general population, we have been working on the construction of allele frequency panels (or reference panels) of Japanese. We have constructed reference panels focusing on single nucleotide variants (SNVs) (Tadaka et al., 2021) and SVs (this work). The former has been used for the design of Japanese-oriented DNA microarrays and genotype imputation of the results. The reference panels have also been used for diagnosis of causative variants of rare diseases (Adachi et al., 2017; Tadaka et al., 2019), as we consider variants that are highly prevalent in the general population (or common variants) are unlikely to be causative variants of rare diseases. Thus, the allele frequency information in the reference panels has been utilized to reduce false positive variants in genome analyses of undiagnosed patients. We believe that this is a fundamental application of the allele frequency panel, which has motivated us to construct a reference panel focusing on SVs in this work. To avoid any confusions, we have revised the sentences as follows:

Line 52: The reference panels have also been used for diagnosis of rare disease, in which the allele frequency information in the general population reduces false positive variants in clinical sequencing analyses of undiagnosed patients (Adachi et al., 2017; Tadaka et al., 2021).

6. l. 65: connection to the next paragraph is broken.

Thank you for the suggestion. We have extensively re-organized the Introduction to

improve the connection between the paragraphs.

7. l. 86: *"has enabled new challenges" sounds strange. I suggest "resulted in (or created) new challenges"*

We changed the sentence as the reviewer mentioned (**Line 87**).

8. l. 102: *Other published strategies (also cited in this work) previously enabled effective construction of population-scale SV panel generation. I suggest using "allows for an effective construction ..."*

We revised the sentence as follows:

Line 108: In the presented study we have succeeded in constructing a population-scale SV panel of Japanese as a fundamental resource for human genetic studies.

9. l. 151-152: *The aim presented is unclear.*

We would like to ask the reviewer to understand the reason why we carried out the Mendelian error profiling. Despite continuous improvements in computational tools, many challenges in read alignment-based SV calling algorithms remain. Therefore, to apply a quality assessment based on Mendelian inheritance error profiling, we designed long-read WGS analyses of 333 participants comprising 111 parent-offspring trios in this study. To clarify this information, we revised the sentence as follows:

Line 174: In this regard, it should be noted that despite the continuous improvements in computational tools, many challenges in read alignment-based SV calling algorithms remain (Jiang et al., 2020; Mahmoud et al., 2019). **Therefore, to apply quality assessments based on Mendelian inheritance error profiling, we designed WGS analyses of 333 BirThree cohort participants comprising 111 parent-offspring trios through** the long-read sequence procedures established in this study (Fig. 2a).

10. l. 153: *"111 parent-offspring trios"*

We reworded the sentence as mentioned (**Line 177**).

11. l. 156-158: *These sentences belong to another section.*

We moved the sentences to the Methods section as follows:

Line 501: Participants. In total, 333 participants composing 111 trios were recruited through the Birth and Three Generation Cohort (BirThree Cohort)(Kuriyama et al., 2020), which was led by the Tohoku Medical Megabank Organization (ToMMo) at Tohoku University and Iwate Tohoku Medical Megabank Organization (IMM) at Iwate Medical University. The participants consisted of almost equal numbers of males (n = 161) and females (n = 172) whose ages ranged from their twenties to eighties (Table 1).

12. l. 163: *Was there a specific reason to use hg37d5 as opposed to a newer reference DNA such as hg38?*

We thank the reviewer for this comment. As most clinical sequence laboratories in Japan still use hg37d5, we employed the older version considering their convenience. However, we certainly understand the reviewer's concern and changed the reference genome to GRCh38. We thoroughly reanalyzed all data by using GRCh38 and presented the new data in this revision. During this revision period, we made both reference panels open to the public.

We would like to inform the reviewer that regarding the characterization of SVs in the Japanese population (Fig. 2d, 2e, 3a, and 3b), the numbers of SVs, size distribution, and distribution of allele frequencies in the GRCh38 version were almost comparable to those in the hg37d5 version. Nonetheless, there are small changes in the numbers of SVs; thus, we revised the related sentences in the main text (**Lines 202, 208, and 240**).

13. l. 156-166: *The authors should also present the sequencing error rates (insertion, deletion, mismatch, and total).*

To address the reviewer's concern, we calculated the sequencing error rate and revised the manuscript as follows:

Line 194: the median sequencing error rate was 7.9% (2.2% for insertions, 3.5% for deletions and 2.2% for mismatches) (Supplementary Fig. 3d).

14. l. 175 *The authors should point to the methods for SV merging.*

We used SURVIVOR (Jeffares et al., 2017) software to merge the SV calls and described the related parameters in the Methods section.

15. l. 176 *Why does it make sense to observe a balanced number of DELs and INSs. Comparison to other large scale long-read SV works needs to be provided here.*

We thank the reviewer for this professional comment. We are also curious about this issue.

The ratio of identified DELs and INSs in previous studies showed more INSs than DELs. We surmise that this discrepancy may be due partly missing or incomplete sequences in the GRCh38 reference genome (Audano et al., 2019; Beyter et al., 2021; Chaisson et al., 2019). Meanwhile, our study shows a balanced number of DELs and INSs. While the precise reason is still uncertain, this observation may be attributable to the difficulties in calling INSs and differences in various criteria used in the detection, merging, and filtering steps of SVs.

To further explore the mechanism underlying this observation, we examined whether some INSs were missing in this study due to bias in the SV calling procedure by introducing a benchmark analysis for the detection of INSs and DELs using the CuteSV algorithm. As we have three assemblies constructed from different individuals, which were constructed from ultradeep sequencing and optical mapping data and exhibit high contiguity and accuracy (Takayama et al., 2021), we decided to use these assemblers. We obtained standard SV-call sets from these assemblies and subjected them to a benchmark analysis (Supplementary Fig 2a). As shown in Supplementary Fig 2b, we observed a slightly lower recall score in INS than DEL using the CuteSV algorithm, indicating that a part of INS may indeed be missing in our pipeline. However, this difference does not explain the magnitude of the difference in INS and DEL between our study and previous studies. Considering these issues, we revised the sentence as follows:

Line 208: Then, we merged these SVs of the 333 individuals into a nonredundant set of SVs to produce a variant repository composed of 37,981 DELs and 36,220 INSs, showing a balanced number of DELs and INSs (Fig. 2e). In this regard, several studies have identified more INSs than DELs (Beyter et al., 2021; Ebert et al., 2021). A plausible explanation for this discrepancy may be the lower recall scores in INS detection than DEL detection in our study (Supplementary Fig. 2b). We surmise that biases in SV calling remain and expect the development of elaborate bioinformatics algorithms.

16. l. 181: I do not see how the data presented supports this claim.

We agree with the reviewer and removed the statement from the manuscript.

17. l. 184: What does it mean "we extracted SVs observed" here? Did you simply use the discovered SVs that are members of a merged SV, or did you genotyped the merged SVs in the individuals?

We would like to ask the reviewer to understand that this is a family-based study, and we had to avoid double counts and SVs in one of the parents and children. To avoid the overestimation of allele frequencies by double counting the variants shared between parents and their son/daughter, we could not utilize all SVs. Therefore, in this study, we selected SVs detected in 222 parents from the variant repository in which all SVs called in 333 individuals exist. To clarify this information, we revised the sentences as follows:

Line 208: Then, we merged these SVs of the 333 individuals into a nonredundant set of SVs to produce a variant repository composed of 37,981 DELs and 36,220 INSs, showing a balanced number of DELs and INSs (Fig. 2e).

Line 234: To avoid double counting the SVs shared between parents and offspring and, thus, prevent the overestimation of the allele frequencies of the SVs in the general population, we extracted SVs observed in 222 unrelated individuals (*i.e.*, fathers and mothers) from the repository to evaluate MAF.

18. l. 191-195: The observation made here requires a proper statistical analysis with hypothesis testing. On the one hand, it is natural to expect large SVs to be more disruptive thus to be rare. On the other hand, false positive SVs may also lead to such findings, and one would expect false positive SVs to tend to appear more as singletons.

We agree with this comment. We conducted a Bonferroni adjusted Wilcoxon rank-sum test to compare the SV size among the MAF categories, which supported the observation that the smallest SVs were found in the common category compared to not only the singleton category but also the rare and low categories. These observations support the notion that the size of SVs appeared to be the smallest in the common category (Line 223 in the revised version). We added asterisks indicating the *P* values to Fig. 3b and revised the related sentence as follows:

Line 833: Distribution of the SV size in each MAF category. **P* = 0.00098 and ***P* < 2×10^{-16} , Holm adjusted Wilcoxon rank sum test.

19. l. 198: Why not insertions as well as deletions?

We agree with this comment. We included INSs in the comparison with the deCODE dataset and obtained concordant results between DELs and INSs. The manuscript was revised as follows:

Line 255: As shown in Figure 3c, of all SVs in our dataset (INS and DEL; shown as “TMM”), 38,304 (53.5%) were also found in the deCODE dataset, while 33,279 were unique to the TMM dataset. Next, we compared the MAFs of the unique

DELs and INs to those of overlapping ones. The results revealed that the SVs in the common category were shared preferentially with those in the deCODE dataset; in contrast, those with a low MAF and those in the rare and singleton categories tended to be unique in the TMM dataset (Fig 3d). Thus, the comparison of the deCODE and TMM datasets revealed significant differences in the ethnic distribution of SVs, even though high-MAF SVs are shared relatively widely across ethnicities.

20. l. 201: the deCODE dataset does report the frequency of the SVs. Please find it in Supplementary Data 2.

We apologize for our oversight. We provided further details in the response to Comment #23 since the comparison between our TMM dataset and the deCODE dataset is heavily expanded in Comment #23.

21. l. 204: There exists a series of SV comparison approaches, used for comparing insertions as well, in the previous works cited in this manuscript. The authors could simply pick one and apply it for insertions as well. Otherwise, the lack of any comparison on roughly half of the merged SV set results in an incomplete analysis.

We agree with this recommendation. By merging the two datasets using the SURVIVOR tool, we evaluated the overlap and correlation of the allele frequencies among the datasets, including INs, as mentioned in the response to Comments #19 and #23.

22. l. 206: This sentence needs a pointer to the methods for SV comparison.

We the described detailed methods as follows:

Line 252: To assess ethnic differences or diversity in the occurrence of SVs, we compared the DELs in our dataset with those in the recently published Iceland deCODE study (Beyter et al., 2021) (*see Methods, “Comparison of SVs to the deCODE dataset”*).

Line 585: Comparison of SVs to published datasets. The SV datasets published by deCODE (Beyter et al., 2021) and gnomAD (Collins et al., 2020) were downloaded from the respective websites (<https://www.nature.com/articles/s41588-021-00865-4> and <https://gnomad.broadinstitute.org>). For the comparison with the gnomAD dataset, we constructed an SV dataset based on hs37d5 reference genome in the same way as the GRCh38 version. The datasets subjected to the comparison were merged

using SURVIVOR software (Jeffares et al., 2017) (version 1.0.6) with the option “1000 1 1 -1 -1 -1”, and the SVs with AF > 0 in each dataset were regarded as detected.

23. 1. 197-212: *I do not think it is useful to compare rare or singleton SVs to the deCODE dataset. I would suggest comparing common SVs. Also this sections, as is, constitutes more of a quality-control rather than providing insight into the differences in ethnic distributions of SVs, does very minimally sheds light on the ethnic diversity of SVs.*

To address this comment, we carried out a comparison of allele frequencies between the deCODE dataset and the TMM dataset utilizing common SVs (MAF \geq 0.05). We found only a weak or moderate correlation between these two datasets (Fig. 3e), suggesting that the differences in the allele frequencies may be due to ethnic differences.

We further expanded the analysis to a comparison to the gnomAD-SV dataset. It should be noted that the gnomAD-SV samples were derived mainly from European and African/African American samples, while less than 10% of the samples were from East Asia. This feature of the gnomAD-SV dataset enables further evaluation to address the ethnic differences in the distribution of allele frequency. In the comparison to the whole gnomAD-SV dataset, we observed only modest correlations between the TMM dataset and the whole gnomAD-SV dataset (Supplementary Fig. 5), which is concordant with the observation in the comparison between the TMM dataset and deCODE dataset. However, when we compared the TMM dataset to the East Asian subset in the gnomAD-SV dataset, much higher correlations were observed between these two datasets (Fig. 3f). This observation indicates that the AFs in the TMM dataset nicely reflect the ethnicity of the AFs in the Japanese population. We revised the manuscript as follows.

Line 264: For further analysis focusing on the ethnic diversity of SVs, we extracted common SVs (MAF \geq 0.05) and compared the allele frequencies (AFs) between the deCODE dataset and the TMM dataset (Fig. 3e). We observed only modest correlations between these two datasets (Pearson correlation coefficient [cor] = 0.50 and 0.40 for DEL and INS, respectively), indicating that the common SVs showed substantial differences in AFs between the ethnicities.

Next, we compared the TMM dataset to that of gnomAD-SV (Collins et al., 2020), which was based on short-read WGS of more than 14,000 samples. Notably, the samples are mainly derived from European and African/African-American samples, while less than 10% of the samples are derived from East Asian samples (EAS). In the comparison between the TMM dataset and the whole gnomAD-SV, only modest correlations (cor = 0.61 and 0.57 for DELs and INSs, respectively)

were observed (Supplementary Fig. 5), showing very good agreement with the observation in the comparison to the deCODE dataset. In contrast, we observed much higher correlations ($\text{cor} = 0.74$ and 0.78 for DELs and INSs, respectively) in the comparison of the TMM dataset and the East Asian subset in the gnomAD-SV (Fig. 3f), demonstrating that the AFs in the TMM dataset reflect ethnicities in AFs.

24. l. 214-251: *This entire section is basically the distribution of SV genotypes in parent-offspring trios, as presented in Fig 4c. This section could be drastically shortened, mainly referring to the figure and mention that the behaviour is comparable to the decode study.*

We agree with the reviewer's comment. The MIE analysis was conducted as a quality control for our SV dataset. We shortened the description of the paragraph as follows:

Line 311: We evaluated the distribution of the SV genotype calls in all parent-offspring trios to assess whether there are specific combinations of genotypes among trios that lead to erroneous calls (Fig. 4c). *While the distributions of DELs and INSs in our dataset were closer to the expected probabilities, for the most part, several minor discrepancies in the genotype distribution were observed. For instance, in the case of the genotype of offspring derived from parents with the genotypes "0/0" and "1/1", high-rate MIE accumulation was observed (Fig. 4c, bottom). This observation was reproducible in the deCODE study (Beyter et al., 2021).* Thus, these results demonstrate that our long-read sequence analysis achieved reasonable genotype calls considering that substantial challenges need to be addressed to accomplish fully reliable and accurate genotype calls of SVs using long-read sequence technology.

25. l. 265-268: *It is well established that SVs are over-represented in centromeric and subtelomeric regions.*

We agree with this comment. We revised the sentence by citing Audano *et al.* paper, which shows concordant results to our study.

Line 327: We also observed several peaks, indicating the high density of SVs on chromosome ends and sites adjacent to the gaps remaining in the reference genome, and *this observation is concordant with the observation in a previous study (Audano et al., 2019) (Fig. 5c).*

26. l. 269-276: *The section that discusses the over and under-representation of SVs within different regions, i.e. intergenic, intronic, exonic, ... etc. requires p-values rejecting the null hypothesis of having no over or under representation.*

We conducted a bootstrap test, and we obtained $P < 0.002$, which rejects the null hypothesis. This result indicates that there is an overrepresentation of SVs in intergenic regions and an underrepresentation of SVs in introns, exons, and CDSs. The manuscript was revised as follows:

Line 342: Of 70,722 SVs, 32,964 (46.6%) were in intergenic regions, while 39,475 (55.8%), 5,413 (7.7%), and 1,823 (2.6%) SVs overlapped with introns, exons, and CDSs, respectively ($P < 0.002$, bootstrap test), which is concordant with a previous study (Beyter et al., 2021).

27. l. 302: *A 32kb deletion in LCE3B also is reported the decode study, associated with psoriasis, which should be to cited in this section.*

We appreciate this comment and added a citation for the deCODE paper.

Line 374: We also found a 32-kb DEL in genes encoding late cornified envelope 3B and 3C (*LCE3B* and *LCE3C*) proteins related to skin barrier functions, as has been described in the previous studies (Beyter et al., 2021; Nestle, Kaplan, & Barker, 2009).

28. l. 317: *As an alternative, the authors can use a long read SV genotyper such as LRcaller, which could provide a better estimate of allele frequency, as opposed to simply using the genotypes from an SV discovery tool.*

We attempted genotyping with LRcaller on our computer server (24 CPU cores, 96 GB memory). However, despite many trials, we could not complete the process due to an insufficient memory error. We surmise that LRcaller might be able to run on a server with much larger memory, but it would be difficult to scale to the large scale of samples. Therefore, we leave genotyping with LRcaller until the algorithm is improved.

29. l. 327: *The discussion section can be substantially shortened.*

We agree with this comment and removed the paragraph concerning the MIE analysis from the discussion since there was a duplicated description in the results section.

Remarks to the Author: Reproducibility:

I mentioned the required further statistical analyses in the section above.
The required parameters for the software being run are provided.

We thank the reviewer for the helpful evaluations. We added the statistical analysis in the revised version as mentioned in the responses to Comments #18 and #26.

Response to Reviewer #3

Overall, the work is carried out to a high standard and will provide a useful resource for the future. I hope the SV database will be accessible and straightforward to access with appropriate permissions.

We appreciate the reviewer for the thorough review, helpful comments, and kind evaluation. We addressed all concerns raised by the reviewer. The reviewer's comments are shown in blue, our responses are shown in black, and the corrected sentences in the text are shown in red. The references are listed at the end of this document.

We agree with this comment and already deposited the SV datasets, which were constructed based on both the hs37d5 and GRCh38 genome references, and uploaded them to our jMorp website (<https://jmorp.megabank.tohoku.ac.jp/202112/>). The website can visualize the SV data through our web-based interface, and users are able to download the data.

1. A major focus of the paper is the use of T-cells to provide DNA. Whilst I think this is an interesting and novel approach I am not sure it is an absolute requirement for sequencing projects of this nature. A comparison with DNA extracted for the same samples for short read sequencing but run on a long read platform would reveal the benefits of the T-cell method. It is important to not give the impression that read N50s of the type generated here are absolutely required for the level of SV detection observed unless this is actually the case. However, the use of the T-cells is an innovative approach.

We thank the reviewer for this professional comment. We would like to inform the reviewer of the advantages of using participant-derived culture cells, such as activated T cells or LCLs, in biobank-based genome analyses. The use of cultured cells as a resource prevents the rapid depletion of the biospecimens stored in the biobank. We surmise that activated T cells and LCLs are both suitable for this purpose. We exploited activated T cells as these cells can be established through much shorter and simpler steps than LCLs; thus, the use of activated T cells as a biological resource has marked benefits for biobank-based analyses. Activated T cells assure the preparation of high-quality DNA samples that support the production of long sequencing reads.

The reviewer raised a concern regarding whether the preparation of high-quality DNA samples utilizing cultured cells is an absolute requirement for long-read sequencing for SV detection. This is an important issue. To ascertain the benefit of preparing high-quality DNA for SV analyses and experimentally answer the reviewer's comment, we assessed the correlation between the read length and SV detection ability. As shown in Fig. 2f, we found a strong correlation between read N50 and the mean size of INs and a

moderate correlation between the N50 and the mean size of DELs. These results demonstrate that SV analyses using longer reads have an advantage in the detection of large SVs over those using shorter reads. We also evaluated the ability to detect SVs with large sizes from 5.9 to 6.1 kb (Supplementary Fig. 4a) and small sizes from 280 to 350 bp (Supplementary Fig. 4b). The SVs belonging to the former fraction involve LINE-1-related SVs, while those in the latter fraction contain Alu-related SVs. As a result, we found that the number of large INSs is correlated with the read N50, whereas that of DELs and small INSs is correlated only marginally. These results clearly indicate that SV analyses using longer libraries, which can be supplied stably by the use of high-molecular weight DNA fragments, are beneficial for the comprehensive detection of large SVs, especially in the case of INSs. Based on these results and observations, we conclude that activated T cells are a good resource as activated T cells can stably supply high-molecular weight genomic DNA. To describe this issue in the revised manuscript, we added the following sentences to the text:

Line 218: To ascertain the benefit of using high-quality DNA in SV analyses, we examined the correlation between the read length and SV detection ability. As shown in Fig. 2f, we observed a strong correlation between the read N50 and the mean size of the INSs and a moderate correlation between the read N50 and the mean size of the DELs (Pearson correlation coefficient [cor] = 0.54 and 0.80 for DEL and INS, respectively). These results suggest that an SV analysis using longer reads has an advantage in the detection of large SVs compared with that using shorter reads. To further verify this finding, we also evaluated the correlation between the read length and SV detection ability by detecting large SVs ranging from 5.9 to 6.1 kb and small SVs ranging from 280 to 350 bp (Supplementary Fig. 4a and b, respectively). The SVs belonging to the former fraction contain LINE-1-related SVs, and those belonging to the latter fraction contain Alu-related SVs. We found that the number of large INSs was correlated with the read N50 (cor = 0.54), whereas that of the DELs (cor = 0.18 and 0.08 for large and small DELs, respectively) and small INSs (cor = 0.25) was not correlated with the read N50. These results indicate that an SV analysis using longer reads utilizing activated T cells is beneficial for the comprehensive detection of large SVs, especially in the case of INSs.

- 2. The paper provides a useful resource for long read technologies and their application but I am concerned that it suggests higher molecular weight DNA is required for a study of this nature than is actually required. Thus the paper in its current form may influence thinking in the field but by reinforcing unsupported views.*

We answered this comment above. We appreciate the reviewer again for raising this important comment. We would like to ask the reviewer to understand that we are presenting an improved method for cohort-based long-read DNA sequencing, but we are not insisting that everybody should use this approach. We believe that this approach for long-read sequencing could be selected depending on the purpose of the studies and that the approach we propose here could among the high-end approaches.

- 3. The authors contend that a limitation on long read sequencing is the requirement for "large amounts of intact DNA". In the following sentence they argue that "In general, the molecular weight of the genomic DNA used for library preparation and its purity substantially affect variant detection, particularly for large SVs, and data yield.". They support this argument with reference to a review from 2019. It would be helpful for the authors to precisely quantify what they mean by a large amount of intact DNA. We routinely run sequencing from as little as 1ug of starting material and the review is not specific in this regard - other users have reported good performance from 300 ng or less. Furthermore, the review doesn't really address the argument the authors make that molecular weight and purity substantially affect variant detection. If the authors are simply distinguishing between "short" and "long" reads then I agree that read length matters - but it is important to be clear on the distinctions. The paper from Beyter et al (long read sequencing from Iceland) has a significantly lower N50 distribution than the work presented here and yet they are also able to analyze SVs.*

Here, the reviewer first commented regarding the actual requirement for a large amount of DNA for long-read sequencing along with the ambiguity of the phrase “a large amount of DNA”. We would like to ask the reviewer to understand that we are presenting an improvement in a method of population-scale genome sequencing. We agree with the reviewer that nanopore sequencing is possible with less than 1 mg of DNA if we operate the process carefully. In contrast, in this study, we wished to produce a structural variant reference panel, and in fact, this reference panel has been open to the public (<https://jmorp.megabank.tohoku.ac.jp/202112/>). We need to pursue stable sequencing procedures. The sample preparation and sequencing workflow of population-scale genomic research with a large number of samples requires robustness to the conditions of each specimen and stability in data production. Therefore, we described that our method used to obtain a relatively large amount of genomic DNA is beneficial for population-scale genomic research using long-read sequencers.

In our study, we used approximately 2 µg of DNA fragments for the library preparation, and this experimental condition is almost comparable to that employed in the deCODE study (Beyter et al., 2021) in which 1-5 µg of DNA were used for the library preparation.

Additionally, the phrase “intact DNA” means high molecular weight DNA. In addition, taking advantage of the stable supply of DNA from activated T cells, our experimental condition is optimized for longer libraries, *i.e.*, more than 25 kb of N50 length, than the deCODE study, which we believe contributes to the detection of longer SVs.

The next comment concerns the relationship between the sequencing read length, which is determined by the molecular weight of the genomic DNA, and SV detection and data yield. First, regarding the relationship between the sequencing read length and downstream SV detection, we already verified this point in the response to Comment #1.

Second, regarding the relationship between the sequencing read length and data yield, we agree that our citation of reference #37 did not support our intent fully and was not adequate. We apologize for the oversight. However, we believe that this statement is supported by the data shown in the deCODE study and our data.

We reanalyzed the Supplementary Data shown in the deCODE study (Beyter *et al.*, 2021) and discovered that the sequencing yields were decreased in unsheared longer-size libraries compared to those in sheared shorter-size libraries (Reviewer’s only Fig. 1a and b). In this revision, we also reanalyzed our data and validated the notion. In our small-scale pilot analysis using the MinION sequencer, the sequencing yields were decreased in the unsheared longer-size library compared with those in the sheared shorter-size library (Reviewer’s only Fig. 1c). These observations indicate that the read length indeed affects the sequencing yield.

The deCODE study also argues that an insufficient sequencing depth can affect downstream variant detection. Therefore, we conclude that it is important to stably supply high-molecular-weight DNA for library preparation for the construction of high-quality reference panels for population-scale genome studies.

Reviewer’s only Fig. 1.

a and b Distribution of read N50 (a) and total coverage (b) per flowcell using sequencing

libraries obtained from sheared (red) and unshered (blue) DNA. The raw data were obtained from Supplementary Data S1 in Bayter *et al.*, *Nat. Genet.* 2021. “Total_coverage” values in the original table were plotted after dividing by 1,000 since there seems to be a mistake in the digits. $*P < 2.2 \times 10^{-16}$, Wilcoxon rank-sum test.

c Relationship between read N50 and sequencing yield per flowcell in our pilot study using the MinION sequencer.

In the revised manuscript, we resolved the ambiguity in our statement regarding the amount of input DNA. We also added an appropriate citation that refers to the relationship between the sequencing read length and data yield.

Line 86: Pioneer studies have taken advantage of such long-read technologies in SV analysis, and the application of this advanced technology to genome analysis **resulted in** new challenges in population-scale studies (Beyter et al., 2021; Wu et al., 2021). In particular, **it is preferable to prepare a relatively abundant amount of high-quality genomic DNA for a human genome analysis using long-read technologies and the design of biobank-based genome research, as the data derived from a long-read sequencing analysis are affected by various factors (Kono & Arakawa, 2019). For instance, the molecular weight of genomic DNA has been shown to affect the sequencing yield (Beyter et al., 2021), which seems to influence downstream variant detection.** These factors should be carefully controlled to avoid false variant detection and genotype calling, which could result in the under- or over-estimation of allele frequencies.

- The authors have a lot of data on the performance of their experiments but it isn't presented. A scatter plot of N50 against yield as well as purity vs yield would support the claims made. In addition, some measure of how the ability to detect SVs varies by read length would be ideal. We have had no problem in generating informative libraries for SV calling from DNA extracted for short read projects. Our read N50s have been in the range of 15-20 kb (similar to that from Beytar et al). I think the authors should demonstrate a link if it exists between the ability to SV call at N50s of 15 kb to 40 kb.*

We thank the reviewer for this comment asking us to provide more basic information related to the performance of each sequencing analysis and more lines of evidence that support the notion that the sequence data yields are negatively correlated to N50. Regarding the relationship between the ability to call SVs and N50, we already addressed this in the response to Comment #1. Regarding the relationship between the purity of DNA and SV detection, we routinely observed this relationship. However, we did not study this issue systematically, and we could not find publications addressing this relationship. Therefore,

we modified the description of the relationship between the purity of DNA and SV detection.

Regarding the sequencing performance, we presented sequencing statistics, including the read N50 length and yield of 430 runs using 411 flowcells (Supplementary Table 1). Among 430 runs, we subjected 323 runs without any problems before and during sequencing for further evaluations (Supplementary Fig. 3). The detailed criteria for the adoption of the data are shown in the Methods. Regarding the relationship between N50 and the sequencing yield, we did not observe a negative correlation (Supplementary Fig. 3c). We surmise that this is because the optimization of the DNA-fragmentation step in our study results in DNA libraries with a lower diversity in N50 length.

To include these data, we revised the manuscript as follows:

Line 180: Using 411 flowcells, we conducted 430 runs in total (Supplementary Table 1).

Line 183: While it has been known that the sequencing yield per flowcell decreases when longer libraries are subjected to sequencing (Beyter et al., 2021), we did not observe such a negative correlation between the two variables (Supplementary Fig. 3c). We surmise that this observation was because optimization in the fragmentation step resulted in a low diversity in the N50 length, suggesting that it is important to optimize the DNA fragmentation step to balance the sequencing yield and read length for high-quality deep sequencing.

Line 540: Sequencing summaries. Among a total of 430 runs, we analyzed 323 runs without any problems before and during sequencing (*i.e.*, we excluded runs for the following reason: 23 runs for difficulty during sequencing; 45 runs using flowcells with a low active pore count below the standard of QC; and 39 runs using flowcells with a lot defect due to manufacturing errors). The sequencing summaries were generated using NanoPlot (De Coster, D'Hert, Schultz, Cruts, & Van Broeckhoven, 2018) software (version 1.27.0).

5. *Pg 7 - lines 135-137 - the authors assert that there is an inverse correlation between library yield and library molecule length (citing the same review). Again, this is a complex statement. Users can obtain high yields (100Gb) on samples prepared with the Nanopore ULK-001 kit for example. Again, the paper contains data that would allow that assessment to be made (plot N50 against yield over many experiments).*

The reviewer noted the inverse correlation between the library yield and library molecule length. We would like to ask the reviewer to understand that we did not make any

statements concerning the “library” yield; rather, we are discussing the “sequence data” yield. We already addressed the relationship between N50 and the sequencing yield in the response to Comment #3 utilizing supporting evidence from the deCODE paper and our pilot study.

While we agree that several scientists have obtained high yields in ultralong sequencing with a Nanopore ULK-001 kit, it should be noted that the sample preparation for this purpose requires careful handling of high-molecular DNA fragments and optimization of the experimental conditions. In the present study, we constructed workflows for the sample preparation and sequencing to analyze several hundreds of samples, which requires robustness to biobank-scale sample processing and stability in data production. Therefore, it is difficult to discuss the sequencing yield in our study in the same context as that in ultralong sequencing.

Nonetheless, we understand the concerns of the reviewer; thus, we revised the sentence to improve the accuracy of the statement.

Line 144: In general, the sequence data yields obtained using a nanopore sequencer widely vary mostly depending on the sample properties, **such as the fragmentation status of the input DNA (Beyter et al., 2021).**

6. *Pg 6. Line 110-11 - I am not sure on the statement "data yield per cost varies significantly dependent on the input.". Yield can be dependent on the quality of sample input, but this could be quantified in some way*

As stated in the response to Comment #3, we intended for the data yield to be dependent on the length of the input library as the deCODE dataset showed. Therefore, we revised our description of the relationship between the quality of the samples and the data yield.

Line 116: Similarly, the data yield per cost significantly varies depending on the input. **For instance, it has been implied that the fragmentation status affects the sequencing yield per run, indicating that the length of the input libraries can affect the sequencing depth per cost (Beyter et al., 2021).**

7. *Pg 7 - line 140-141. This sentence is not clear. Are the yields for reads greater than 20Kb? "More than 20kb of N50" is unclear. Figure 1e doesn't show anything about yield, rather it is showing read length against quality with a heat map of read count. Converting this to base pairs is somewhat complicated.*

This comment overlaps and is repetitive with Comment #11. To further substantiate the conclusion "More than 20 kb of N50", we added detailed data indicating the sequence yield per flowcell and read N50 obtained from the genomic DNA samples shown in Fig. 1d (n=5) in Supplementary Fig. 1 and revised the sentence as follows:

Line 149: We obtained 85.0 ± 5.4 Gb of yield and 25.8 ± 1.8 kb of N50 length per flowcell (n = 5), indicating that half of the sequence base pairs were derived from reads longer than or equal to 25.8 kb (Fig. 1e and Supplementary Fig. 1a and 1b).

8. *Pg 10 - lines 230-232 - can the authors elaborate on which technical improvements they think have caused the change?*

We surmise that the factors that contribute to the improvement in MIE profiles include stable data production in terms of sequencing depth and read length, as well as in bioinformatics pipelines. However, the precise reasons remain to be clarified.

Line 295: Thus, although the MIE rates were still higher than expected, there was a substantial improvement in concordance with Mendelian inheritance perhaps due to technical improvements in long-read sequencing, including stable data production in terms of the sequencing depth and read length, and in bioinformatics pipelines.

9. *Pg 12 - lines 259 - The authors show that SVs occur in close proximity to gaps in the reference and imply this is an artefact - to address this they simply omit these regions from their analysis. This led me to ask which reference they used. Close inspection reveals the authors used the hs37d5 reference based on the 1000 genomes project but not a later more complete assembly such as hg38 or a more recent telomere-to-telomere assembly. Obviously changing the entire study to a new reference is significant, but I wonder why the authors chose the earlier assembly base? They resolve the ambiguity of SVs in close proximity to gaps by removal from the analysis (if I understand correctly) but it would be interesting to determine if mapping to more contiguous reference resolved this issue.*

We agree with the reviewer and reanalyzed all data in the revised version using GRCh38.

Regarding the distribution of SVs, we observed that SVs also occur near gaps in GRCh38 and hs37d5. We did not omit the SVs close to gaps from the analyses.

10. *Pg 20 - line 456 - I think the authors are referencing a mean Phred score > 6 - and so this should be described as a mean quality score? (i.e Nanopore "Pass" reads?).*

We thank the reviewer for this professional comment and have corrected the wording as the reviewer mentioned (**Lines 191, 549, and 814**).

11. Figure 1e - I'm not convinced this panel clearly presents the statements the authors make about it in the text (see 7).

As described in the response to Comment #7, we added data indicating the sequence yield per flowcell and read N50 obtained from the genomic DNA samples shown in Fig. 1d to Supplementary Fig. 1a and 1b, respectively.

12. Figure 2 - a scatter plot of N50 vs yield would be interesting (but should also include some reference to pore count on the flowcell).

Finally, we thank the reviewer for raising several questions regarding the relationship between N50 and the sequencing yield. We addressed this point mainly in the response to Comments #3, #5 and #6. We believe that these responses and the resulting revisions in the text, including the addition of new Supplemental Figures, substantially improved this paper.

Regarding the pore count, we evaluated the relationship between the active pore count of the flowcell and the sequencing yield (Supplementary Fig. 3a) and the N50 length and the sequencing yields (Supplementary Fig. 3b). The sequencing yields were increased with an active pore count, but the N50 lengths were not. These observations indicate that the quality of the flowcell was a determinant of the sequencing yield.

Line 180: As shown in Supplementary Fig. 3a and 3b, the sequencing yields increased as the active pore count increased, but the N50 lengths did not. These observations indicate that the quality of the flowcell is a determinant of the sequencing yield.

Reference

- Adachi, T., Kawamura, K., Furusawa, Y., Nishizaki, Y., Imanishi, N., Umehara, S., . . . Suematsu, M. (2017). Japan's initiative on rare and undiagnosed diseases (IRUD): towards an end to the diagnostic odyssey. *Eur J Hum Genet*, *25*(9), 1025-1028. doi:10.1038/ejhg.2017.106
- Audano, P. A., Sulovari, A., Graves-Lindsay, T. A., Cantsilieris, S., Sorensen, M., Welch, A. E., . . . Eichler, E. E. (2019). Characterizing the Major Structural Variant Alleles of the Human Genome. *Cell*, *176*(3), 663-675 e619. doi:10.1016/j.cell.2018.12.019
- Beyter, D., Ingimundardottir, H., Oddsson, A., Eggertsson, H. P., Bjornsson, E., Jonsson, H., . . . Stefansson, K. (2021). Long-read sequencing of 3,622 Icelanders provides insight into the role of structural variants in human diseases and other traits. *Nat Genet*. doi:10.1038/s41588-021-00865-4
- Bolognini, D., & Magi, A. (2021). Evaluation of Germline Structural Variant Calling Methods for Nanopore Sequencing Data. *Front Genet*, *12*, 761791. doi:10.3389/fgene.2021.761791
- Chaisson, M. J. P., Sanders, A. D., Zhao, X., Malhotra, A., Porubsky, D., Rausch, T., . . . Lee, C. (2019). Multi-platform discovery of haplotype-resolved structural variation in human genomes. *Nat Commun*, *10*(1), 1784. doi:10.1038/s41467-018-08148-z
- Collins, R. L., Brand, H., Karczewski, K. J., Zhao, X., Alfoldi, J., Francioli, L. C., . . . Talkowski, M. E. (2020). A structural variation reference for medical and population genetics. *Nature*, *581*(7809), 444-451. doi:10.1038/s41586-020-2287-8
- De Coster, W., D'Hert, S., Schultz, D. T., Cruys, M., & Van Broeckhoven, C. (2018). NanoPack: visualizing and processing long-read sequencing data. *Bioinformatics*, *34*(15), 2666-2669. doi:10.1093/bioinformatics/bty149
- Ebert, P., Audano, P. A., Zhu, Q., Rodriguez-Martin, B., Porubsky, D., Bonder, M. J., . . . Eichler, E. E. (2021). Haplotype-resolved diverse human genomes and integrated analysis of structural variation. *Science*, *372*(6537). doi:10.1126/science.abf7117
- Holley, G., Beyter, D., Ingimundardottir, H., Moller, P. L., Kristmundsdottir, S., Eggertsson, H. P., & Halldorsson, B. V. (2021). Ratosk: hybrid error correction of long reads enables accurate variant calling and assembly. *Genome Biol*, *22*(1), 28. doi:10.1186/s13059-020-02244-4
- Jeffares, D. C., Jolly, C., Hoti, M., Speed, D., Shaw, L., Rallis, C., . . . Sedlazeck, F. J. (2017). Transient structural variations have strong effects on quantitative traits and reproductive isolation in fission yeast. *Nat Commun*, *8*, 14061. doi:10.1038/ncomms14061
- Jiang, T., Liu, Y., Jiang, Y., Li, J., Gao, Y., Cui, Z., . . . Wang, Y. (2020). Long-read-based human genomic structural variation detection with cuteSV. *Genome Biology*, *21*(1). doi:10.1186/s13059-020-02107-y

- Kono, N., & Arakawa, K. (2019). Nanopore sequencing: Review of potential applications in functional genomics. *Dev Growth Differ*, *61*(5), 316-326. doi:10.1111/dgd.12608
- Kourkouta, E., Weij, R., Gonzalez-Barriga, A., Mulder, M., Verheul, R., Bosgra, S., . . . Datson, N. A. (2019). Suppression of Mutant Protein Expression in SCA3 and SCA1 Mice Using a CAG Repeat-Targeting Antisense Oligonucleotide. *Mol Ther Nucleic Acids*, *17*, 601-614. doi:10.1016/j.omtn.2019.07.004
- Kuriyama, S., Metoki, H., Kikuya, M., Obara, T., Ishikuro, M., Yamanaka, C., . . . Yamamoto, M. (2020). Cohort Profile: Tohoku Medical Megabank Project Birth and Three-Generation Cohort Study (TMM BirThree Cohort Study): rationale, progress and perspective. *Int J Epidemiol*, *49*(1), 18-19m. doi:10.1093/ije/dyz169
- Mahmoud, M., Gobet, N., Cruz-Davalos, D. I., Mounier, N., Dessimoz, C., & Sedlazeck, F. J. (2019). Structural variant calling: the long and the short of it. *Genome Biol*, *20*(1), 246. doi:10.1186/s13059-019-1828-7
- Mitsuhashi, S., Frith, M. C., & Matsumoto, N. (2021). Genome-wide survey of tandem repeats by nanopore sequencing shows that disease-associated repeats are more polymorphic in the general population. *BMC Med Genomics*, *14*(1), 17. doi:10.1186/s12920-020-00853-3
- Nestle, F. O., Kaplan, D. H., & Barker, J. (2009). Psoriasis. *N Engl J Med*, *361*(5), 496-509. doi:10.1056/NEJMra0804595
- Sedlazeck, F. J., Rescheneder, P., Smolka, M., Fang, H., Nattestad, M., von Haeseler, A., & Schatz, M. (2017). Accurate detection of complex structural variations using single molecule sequencing. doi:10.1101/169557
- Tadaka, S., Hishinuma, E., Komaki, S., Motoike, I. N., Kawashima, J., Saigusa, D., . . . Kinoshita, K. (2021). jMorp updates in 2020: large enhancement of multi-omics data resources on the general Japanese population. *Nucleic Acids Res*, *49*(D1), D536-D544. doi:10.1093/nar/gkaa1034
- Tadaka, S., Katsuoka, F., Ueki, M., Kojima, K., Makino, S., Saito, S., . . . Kinoshita, K. (2019). 3.5KJPNv2: an allele frequency panel of 3552 Japanese individuals including the X chromosome. *Hum Genome Var*, *6*, 28. doi:10.1038/s41439-019-0059-5
- Takayama, J., Tadaka, S., Yano, K., Katsuoka, F., Gocho, C., Funayama, T., . . . Tamiya, G. (2021). Construction and integration of three de novo Japanese human genome assemblies toward a population-specific reference. *Nature Communications*, *12*(1). doi:10.1038/s41467-020-20146-8
- Wang, J. R., Holt, J., McMillan, L., & Jones, C. D. (2018). FMLRC: Hybrid long read error correction using an FM-index. *BMC Bioinformatics*, *19*(1), 50. doi:10.1186/s12859-018-2051-3

Wu, Z., Jiang, Z., Li, T., Xie, C., Zhao, L., Yang, J., . . . Xie, Z. (2021). Structural variants in the Chinese population and their impact on phenotypes, diseases and population adaptation. *Nat Commun*, 12(1), 6501. doi:10.1038/s41467-021-26856-x

Response to the Reviewers

Response to Reviewer #3

The authors have addressed the significant majority of my comments to my satisfaction.

I agree with the authors that having a consistent and robust source of input DNA to a sequencing experiment is vital. My concern is do you need molecules with a mean length of 10kb or 100kb? In my view, the paper does not provide evidence that molecules (and thus reads) of >100kb are required for insights into SVs - and my point is that many studies have collected DNA that falls at the lower end of this range and may be perfectly suitable for sequencing. I think the text now reflects this view.

We appreciate the reviewer for the understanding of the importance of a consistent and robust source of input DNA. The reviewer concerns that molecules and reads with a length of >100kb are absolutely needed for the SV analysis. As the reviewer mentioned in the above comment, we are not insisting or have not provided evidence to suggest the requirement of molecules or reads with a length of >100kb. We observed that the read length ranging from 10 kb to 35 kb is correlated with the ability of SVs detection, especially in the case of INs (paragraph starting from Line 223). We surmise that this limitation is due to the sample preparation procedure in this study. To balance the sequencing yield and read length, in this study, we adopted DNA fragments with lengths ranging from 20 kb to 80 kb for the library preparation. To improve the clarity, we have revised the description (Line 236)

Response to Reviewer #4

This work is significant because it reflects the second large scale long-read sequencing of a specific population. If the data are shared, it will enable additional studies of rare variation discovered by long reads.

Claims are mostly discussed in the context of previous literature. One example of this not being the case is given below.

We appreciate the reviewer for the thorough review, evaluation, and constructive comments. We addressed all the concerns raised and have revised the manuscript along the lines suggested by the reviewer.

- I agree with the authors that a single SV discovery pipeline simplifies analysis. Furthermore, one would have to determine how to merge different calls, and the standard outcomes of an intersection improving precision and union improving recall would likely apply.*

We thank the reviewer for the understanding of the benefits of simplifying the SV analysis using a single SV discovery pipeline. We agree with the comment. As the reviewer mentioned, when multiple pipelines used, results of SV analysis may be influenced by the method used to merge different calls. We have evaluated the effect of the methods merging different calls utilizing our benchmark dataset. As expected, the intersection of SV calls from CuteSV and sniffles improved the precision at the expense of recall, while the union of them improved the recall at the expense of precision. (Reviewer's only Fig. 1). To avoid influence of the method merging the different calls and making the interpretation not complicated, we used single SV discovery pipeline in this study.

Reviewer's only Figure 1. Precision-recall plot. The methods merging results are represented by color, whereas the samples are represented by shapes as specified in the legend.

- The second reviewer brought up the discrepancy of the ratio between insertion and deletion calls in this study compared to previous studies. The explanation offered, that there is a lower recall for insertion SV than deletion may not be sufficient. The recall drops off at longer SV, but most SV called in long read studies are < 2kb.*

One additional hypothesis is that a few thousand loci are under-represented in GRCh38 (too short), and are expanded in most individuals. For small sample sizes (Ebert et al), this may tip the balance to insertions. One could show the ratio of insertion to deletion as additional genomes are added, and it should get closer to 50:50 as the population size increases, then likely go up again as more rare insertions due to MEI are detected.

We thank the reviewer for the professional comment. The reviewer provided a fascinating hypothesis that a few thousand loci under-represented in GRCh38 may affect the ratio between insertions and deletions. We have re-evaluated our data and observed data which support the hypothesis.

In the analysis for individual (Fig. 2d), we identified more insertions than deletions (12,133 and 10,923 per individual, respectively). The result may indicate that the loci under-represented in GRCh38 are detected as insertions that everyone harbors, which resulted in the observation that more insertions than deletions are identified at individual-level (or small scale) analysis. When the analysis is expanded to the population scale that contains 333 individuals (Fig. 2e), the ratio between insertions and deletions gets closer to 50:50 (36,220 and 37,981, respectively). This observation seems to support the hypothesis that the expansion of population size and identification of SVs with lower allele frequency attenuates the insertion-bias. We have added the observation and the hypothesis to the text (Line 214).

- The callset should be formed from either unrelated individuals, or taking care to ensure that the ~3% of calls from children that violate Mendelian inheritance are handled appropriately. If we trust the callset, the total number of calls should be similar on parents alone, and children offer a great chance to validate ~66% of calls (homozygous + inherited het).*

We agreed with the reviewer comment. We calculated MAFs and constructed the dataset from 222 unrelated individuals (*i.e.*, fathers and mothers) (paragraph starting from Line 239). In addition, following to the comment, we have carried out evaluations of the Mendelian inheritance for all SVs by comparing the offspring's genotypes and the parents' genotypes.

Among 74,201 SVs observed either in parents or in offspring, we found that 1,731 SVs (2%) were not observed in the parents but were observed in the offspring. In other words, the 2% of the SVs were homozygotes (0/0) of reference allele in all parents, but they were either homo- or

heterozygote of alternate allele (1/1 or 0/1, respectively) in the offspring. When we trust the callset from the parents, the 1,731 SVs can be regarded as those violating Mendelian inheritance. We further expanded the analysis including all combinations of the genotypes observed in the parent-offspring trios. We calculated the error family ratio (number of trios with MIE/total number of trios analyzed) to evaluate the incidence of the MIEs of each SV. As shown in Supplementary Fig. 6a and b, we observed that the error family ratios are fundamentally low, but the ratio were relatively high in case of the SVs located near gaps and chromosome ends. We have included these observations in our manuscript (paragraph starting from Line 308). In addition, we included the error family ratios for each SV into our datasets and made it publicly available from our website (Line 614).

4. *Lines 244: allele frequency decreasing with size has been noted for some time (Cooper, G. et al, Nat. Gen, 2011), though this can confirm the results extend to smaller SV sizes.*

We appreciate for the professional comment. As the reviewer mentioned, Cooper *et al* analyzed copy number variations with size from approximately 100 kb to 1 Mb and described the allele frequency decreasing with size. Our result can extend the observation to smaller SV with size of several hundred base-pairs. We added the description to the manuscript including the citation of Cooper *et al*, *Nat. Genetics*, 2011 (Line 253).

5. *More effort should be taken to ensure accuracy of the SVs that are singletons and low frequency in the callset. You can remark on the number of low-frequency SVs that are validated by inheritance.*

We appreciate the reviewer for the professional comment. Following this advice, we have evaluated accuracy of the SVs, which are singletons with low frequency, by employing the value of Mendelian error family ratio (see the response to the comment #3). We have presented the data in the Supplementary Fig. 6c. As can be seen, the Mendelian error family ratios are lower in singleton SVs than in common or highly frequent SVs. This is against our expectation and rather a surprise. We surmise one plausible explanation for this observation is that the accuracy of the SVs with high frequency may be affected by systematic errors during the detection of the SVs. In low complexity regions in the genome, some of the SVs with low accuracy may be called with high frequencies, which results in Mendelian inheritance errors. In contrast, singletons and low-frequency SVs may be less affected by such systematic errors. We added this discussion to the manuscript (paragraph starting from Line 319 and figure legend starting from Line 946).

Minor Comments

1. *Section: SVs associated with clinical phenotypes. DEL and INS are used, but t*

We could not take meaning of this comment.

2. *The methods are reproducible, though it does not seem the raw data is accessible.*

We appreciate for the evaluation. The individual sequence data and genotyping results based on which the allele frequency datasets were constructed are available upon request after approval by the Ethical Committee and the Materials and Information Distribution Review Committee of ToMMo (Line 616).